# Bilateral equalization of synaptic output in olfactory glomeruli of *Xenopus* tadpoles

**Marta Casas[1,2], Beatrice Terni[1,2], Artur Llobet[1,2]\***

[1]Laboratory of Neurobiology, Department of Pathology and Experimental Therapy, Institute of Neurosciences, University of Barcelona, L'Hospitalet de Llobregat, Spain; [2]Bellvitge Biomedical Research Institute (IDIBELL), L'Hospitalet de Llobregat, Spain

## eLife Assessment

This manuscript investigates inter-hemispheric interactions in the olfactory system of Xenopus tadpoles. Using a combination of electrophysiology, pharmacology, imaging, and uncaging, the transection of the contralateral nerve is shown to lead to larger odor responses in the un-manipulated hemisphere, and implicates dopamine signaling, likely originating from the lateral pallium, in this process. The study **convincingly** uses a rich and sophisticated array of tools to investigate olfactory coding, and uncovers **valuable** mechanisms of signaling likely to be conserved across vertebrates.

**\*For correspondence:**
allobet@ub.edu

**Competing interest:** The authors declare that no competing interests exist.

**Abstract** Odorants stimulate olfactory sensory neurons (OSNs) to create a bilateral sensory map defined by a set of glomeruli present in the left and right olfactory bulbs. Using *Xenopus tropicalis* tadpoles, we challenged the notion that glomerular activation is exclusively determined ipsilaterally. Glomerular responses evoked by unilateral stimulation were potentiated following transection of the contralateral olfactory nerve. The gain of function was observed as early as 2 hr after injury and faded away with a time constant of 4 days. Potentiation was mediated by the presence of larger and faster calcium transients driving glutamate release from OSN axon terminals. The cause was the reduction of the tonic presynaptic inhibition exerted by dopamine $D_2$ receptors. Inflammatory mediators generated by injury were not involved. These findings reveal the presence of a bilateral modulation of glomerular output driven by dopamine that compensates for imbalances in the number of operative OSNs present in the two olfactory epithelia. Considering that the constant turnover of OSNs is an evolutionarily conserved feature of the olfactory system and determines the innervation of glomeruli, the compensatory mechanism described here may represent a general property of the vertebrate olfactory system to establish an odor map.

## Introduction

Olfactory glomeruli are spherical regions of neuropil that contain the first synapse for processing olfactory information and form the glomerular layer of the olfactory bulb. Despite variations in number or size among vertebrate species, glomeruli show an evolutionarily conserved synaptic connectivity (**Shepherd et al., 2004**). Glomerular input comes from axon terminals of olfactory sensory neurons (OSNs) expressing the same odorant receptor that transfer information to the dendrites of mitral cells. Synaptic transmission between OSNs and mitral cells is regulated by inhibitory and excitatory contacts established with juxtaglomerular neurons. This connectivity allows that upon exposure to odorants, only a defined set of glomeruli is activated, creating an odor map (**Mombaerts et al., 1996**). Glomeruli exhibit an anatomical and functional symmetrical distribution in the left and right olfactory bulbs;

therefore, the sensory map created in the olfactory bulb is bilateral and is generated by homologous glomeruli reflecting the contribution of the two nasal cavities (*Lodovichi, 2021*; *Belluscio and Katz, 2001*).

OSNs originate in the olfactory epithelium and are subject to a continuous turnover throughout life (*Holl, 2018*). An efficient neuronal replacement is accomplished by the rapid connection of newborn OSNs to the olfactory bulb. For example, in *X. tropicalis* tadpoles, OSNs can establish functional synapses in a time window of 4 days (*Terni et al., 2017*), and, in mice, it takes a week for newborn OSNs to get inserted in the olfactory bulb through a *plug-and-play* mechanism (*Browne et al., 2022*). The balance existing between the elimination of OSNs and their insertion in olfactory bulb circuitry likely determines a range of input neurons innervating a glomerulus, rather than a constant, precise figure. In this context, how the formation of an odor map accounts for possible variations in the number of input neurons remains unknown. Considering that OSNs exclusively form ipsilateral synapses (*Shepherd et al., 2004*) and release glutamate with near-maximal release probability (*Murphy et al., 2004*), the possible crosstalk balancing the individual contribution of bilaterally distributed glomeruli remains unknown because their high output gain is assumed to be defined unilaterally.

Here, we take advantage of the experimental capacities offered by *Xenopus tropicalis* tadpoles to assess whether the output of a genetically labeled glomerulus (*Terni et al., 2017*; *Terni and Llobet, 2021*) is solely determined by ipsilateral stimulation. In vivo recordings carried out after disrupting the contribution of the contralateral pathway challenged this notion. Our results show that the tonic inhibition of glomerular responses exerted by dopamine $D_2$ receptors is bilaterally shaped to compensate for differences in the number of OSNs innervating the contralateral olfactory bulb. Considering the evolutionarily conserved developmental, morphological, and functional features of the *Xenopus* tadpole olfactory system (*Menini, 2010*; *Manzini et al., 2022*), the homeostatic mechanism here described might represent a general rule for the formation of an integrated odor map in vertebrates.

## Results
### Characterization of odor-evoked responses in a genetically defined glomerulus

The *X. tropicalis* line *Dre.mxn1:GFP* allows the identification of an olfactory glomerulus located laterally and innervated by a discrete population of OSNs (*Terni et al., 2017*; *Terni and Llobet, 2021*). We targeted an electrode to the left GFP-labeled glomerulus and recorded changes in the local field potential (LFP) evoked by ipsilateral stimulation with an amino acid acting as a waterborne odorant (*Figure 1A and B*). An olfactory response was triggered by a 100 ms application of 200 µM methionine to the olfactory epithelium, thus confirming the ability of *Xenopus* tadpoles to detect amino acids (*Manzini et al., 2007*). Stimulation evoked a negative deflection of the LFP that resembled responses obtained in the glomerular layer of rats upon sniffing odors (*Chaigneau et al., 2007*). A main difference between negativities recorded in *X. tropicalis* and those found in rodents is that the latter are respiration-locked, while in tadpoles there was a single deflection that recovered with a half-time ranging from 1 s to 4 s.

An estimate of the spatial extent of the LFP signal was obtained by evaluating responses after changing the position of the recording electrode. The characteristic profile of LFP changes disappeared when the pipette was directed to the mitral cell layer (*Figure 1C*), thus illustrating they were confined to the glomerular layer. The consistent success rate (85%, n=269) obtained by targeting the GFP-labeled glomerulus with the recording electrode decreased to 50% (n=36) when random locations linearly spaced by 50 µm were tested within the glomerular layer. The example shown in *Figure 1D* illustrates how the characteristic response to methionine was obtained only in one out of four positions tested. These observations could be explained by the capacity of the electrode to detect activated glomeruli (*Manzini et al., 2007*). The high success rate of the recordings guided by fluorescence supports that the GFP-labeled glomerulus responded to methionine. Considering many olfactory glomeruli in *Xenopus* tadpoles are broadly tuned (*Manzini et al., 2007*), it is conceivable that the recorded glomerulus was also activated by other waterborne odorants, which were not tested in the current study.

The LFP signal likely sampled the entire volume of 15765±2119 µm³ (n=33) defined by fluorescence but, as the lateral cluster of glomeruli is markedly involved in the detection of amino acids

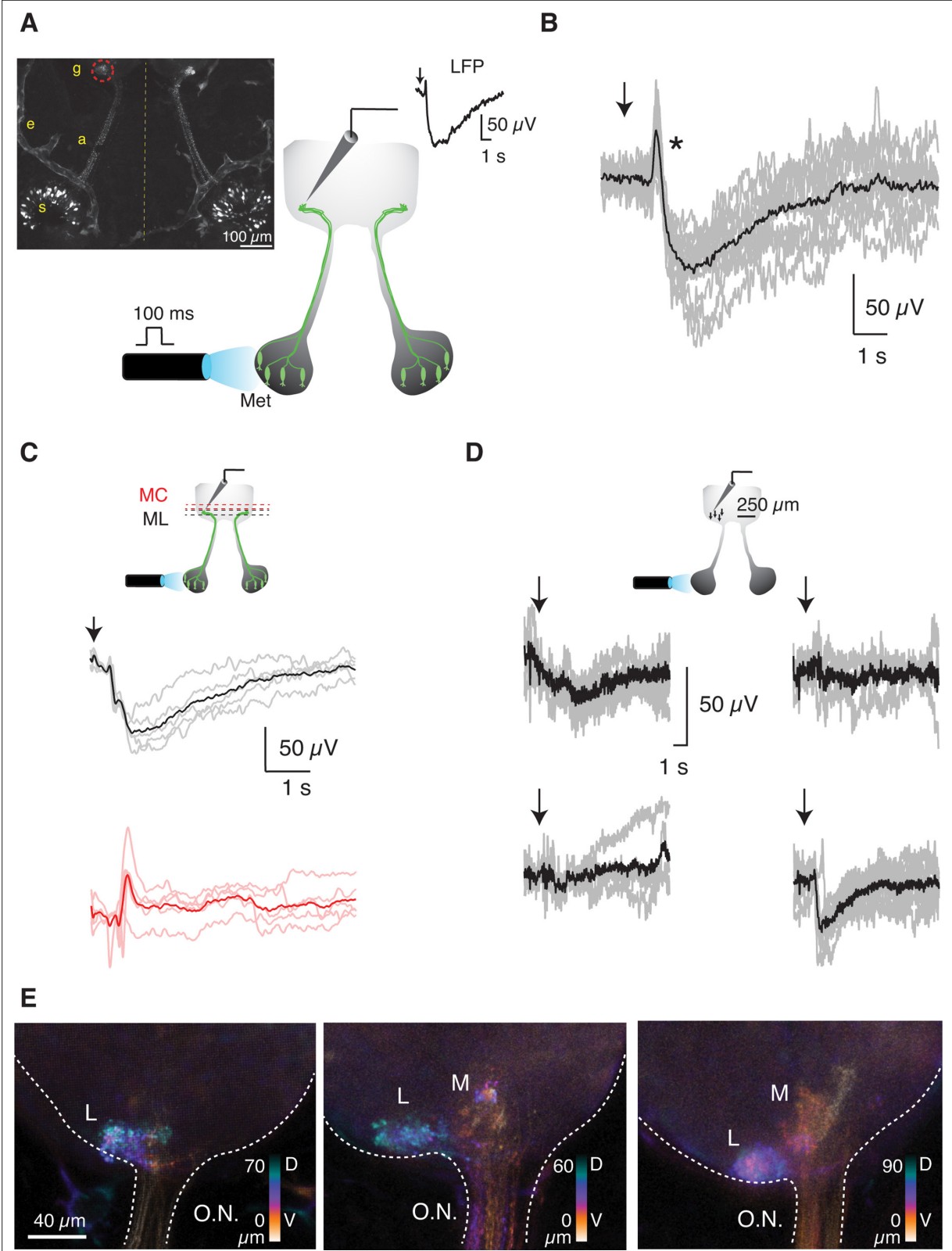

**Figure 1.** Recording of odor-evoked responses in a genetically defined glomerulus. (**A**) Schematic diagram illustrating the experimental preparation. A puff of 200 μM methionine was applied for 100 ms to the left olfactory epithelium of *Dre.mxn1:GFP X. tropicalis* tadpoles. Changes in the local field potential (LFP) were measured ipsilaterally with an electrode targeted to the lateral glomerulus formed by axon terminals of olfactory sensory neurons (OSNs) expressing GFP. The inset shows a confocal projection illustrating labeled OSNs (s: soma, a: axon) and the bilateral formation of

*Figure 1 continued on next page*

*Figure 1 continued*

glomeruli (g). The *Dre.mxn1* promoter also drives GFP expression in the endothelial cells of some blood vessels (e). (**B**) Representative glomerular odor-evoked response (black) obtained by averaging individual responses (gray) of the LFP following stimulation (arrow). The asterisk shows a positivity that was evident in 37% of the recordings and preceded the characteristic negativity associated with glomerular activation. (**C**) Negative deflections of LFP (individual responses, gray; average trace, black) were observed when the recording electrode was placed in the GFP-labeled glomerulus but disappeared in the mitral cell layer (ML, red traces; GL, glomerular layer). (**D**) Experiment showing LFP recordings performed in four different locations of the glomerular layer spaced by 50 μm. The characteristic odor-evoked response was observed in only one of the positions tested. The representative glomerular odor-evoked response (black) was obtained by averaging individual responses (gray) following stimulation with methionine (arrow). (**E**) Hyperstack projections of the left olfactory bulb of three different *Dre.mxn1:GFP X. tropicalis* tadpoles. The lateral glomerulus (L) is always evident, and some medial projections (M) are apparent in two of the illustrated examples. The color scale indicates dorsoventral disposition.

(*Weiss et al., 2021a*), the influence of adjacent glomeruli could not be ruled out from the start. Considering the labeled glomerulus had a diameter of ~30 μm (*Figure 1E*) and there was a 50 μm axial resolution to obtain independent readouts (*Figure 1D*), only a portion of surrounding glomeruli could theoretically participate in the observed LFP signal. The glomerular layer of the amphibian olfactory bulb is loosely organized compared to mammals, as glomerular units show different sizes and lack ensheathing astrocytes (*Nezlin and Schild, 2000*; *Gaudin and Gascuel, 2005*). The absence of a homogeneous distribution of glomeruli, added to the dorsorostral location of GFP-labeled axon terminals (*Figure 1E*), restricted the number of adjacent glomeruli that could theoretically contribute to the recorded LFP signal to those found ventrally or caudally. Even if 50% of such units were activated by methionine, the decay of the LFP following a relationship inversely related to distance to the recording site, as reported for the mammalian olfactory bulb (*Karnup et al., 2006*), would minimize their contribution. Together, these results indicate that methionine application to the olfactory epithelium reliably activated the lateral glomerulus labeled by GFP in the *Dre.mxn1:GFP* line.

Glomerular responses were assayed by repetitive stimulation at 2 min intervals, and all negativities showed a comparable profile (*Figure 1B*). The peak of the negative deflection was reached ~1 s after olfactory stimulation. The characteristic LFP response of any given tadpole was obtained by averaging 5–12 consecutive stimulations. Local application of 100 μM CNQX decreased LFP negativities by approximately 50% (*Figure 2A*). The reduction was comparable when 100 μM D-AP5 was used instead of CNQX (*Figure 2B*). The concomitant application of both drugs produced an additive effect that reduced responses by 70% (*Figure 2C*, p=0.0004, paired t-test), thus evidencing their synaptic origin, as well as the involvement of AMPA and NMDA receptors. These results are comparable to those obtained in rats (*Chaigneau et al., 2007*; *Lecoq et al., 2009*), which validates our experimental configuration to record LFP responses in a region defined by a genetically labeled glomerulus.

The onset phase of negativities was particularly sensitive to CNQX and AP5, suggesting that it reflected the involvement of receptors activated by glutamate release from OSN axon terminals. This possibility was tested by inducing a jump in glutamate concentration inside the glomerulus. We targeted the electrode to the lateral GFP glomerulus, injected *Xenopus* Ringer solution containing 300 μM Rubi-glutamate, and uncaged it using a flash of blue light (*Fino et al., 2009*). A transient negativity was induced by a 500 ms pulse of light, which was reproduced by repeated stimulations (*Figure 2D*). The average amplitude of $LFP_{peaks}$ was 30±3 μV (n=21, 3 tadpoles). This observation supported that the initial phase of LFP negativities was caused by glutamate secreted by OSN axon terminals that converged in a glomerulus.

The contribution of inhibitory neurotransmission mediated by $GABA_A$ receptors was negligible because no variations in LFP changes evoked by methionine were observed after local application of 1 mM picrotoxin (*Figure 2E*). To verify that ipsilateral stimulation of OSNs was the trigger of recorded responses, we investigated the effect of methionine application to the contralateral epithelium. Negativities disappeared (*Figure 2F*), thus demonstrating that glomerular activation was exclusively unilateral. In 37% of the tadpoles studied (n=174), the negativity was preceded by a transient positivity (*Figure 1B*). When present, this LFP deflection was not modified by glutamate and $GABA_A$ receptor antagonists (*Figure 2A and E*), thus indicating it was unrelated to synaptic mechanisms. As the positivity was caused by the application of olfactory stimuli and always occurred before the synaptic response, it is possible to consider its origin in cellular structures conveying peripheral information to the olfactory bulb, such as the layer of nerve fibers.

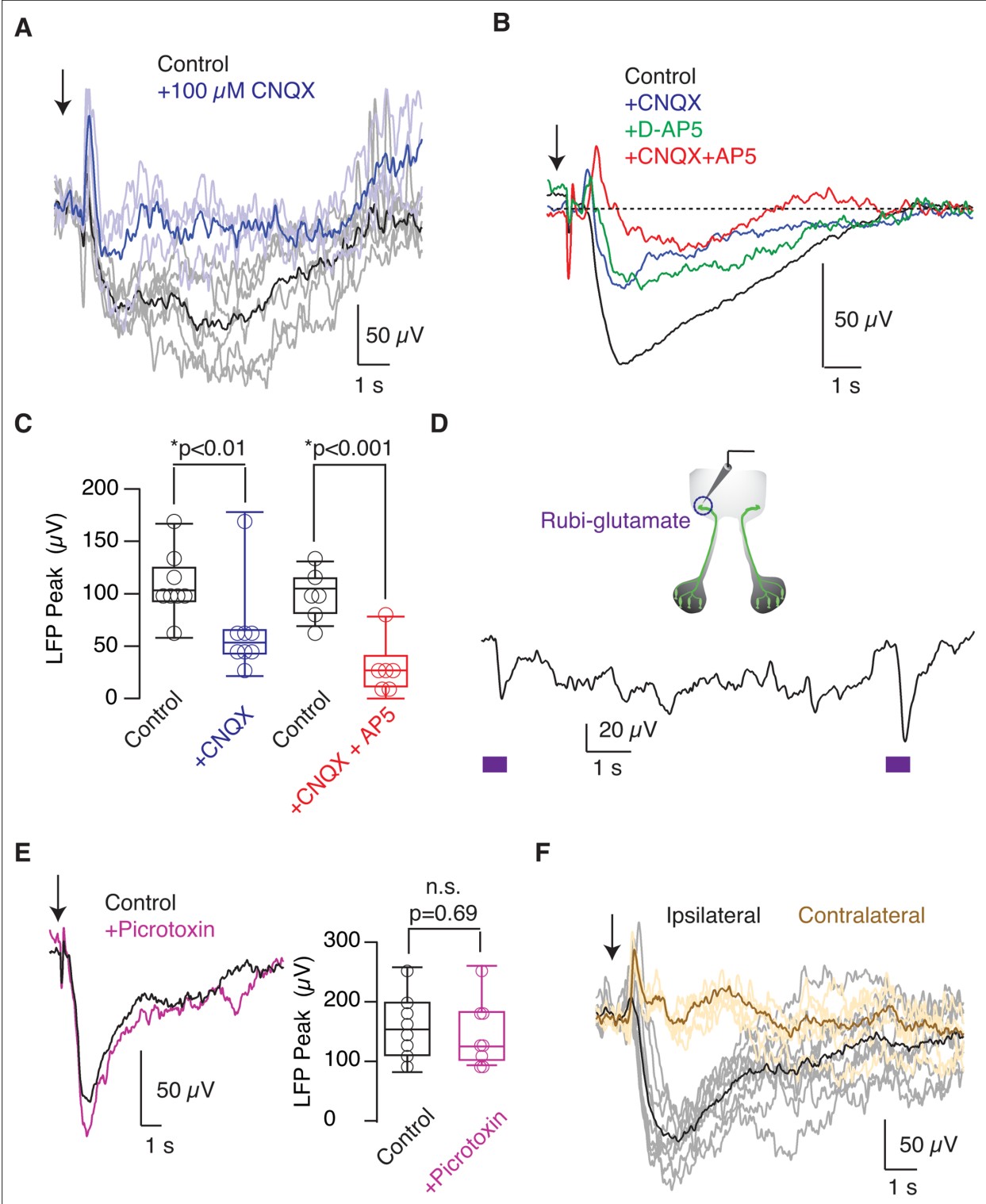

**Figure 2.** Odor-evoked responses are mediated by glutamatergic neurotransmission. (**A**) The representative glomerular odor-evoked response (black) was obtained by averaging individual responses (gray) of the local field potential (LFP) following ipsilateral stimulation of the olfactory epithelium using 200 µM methionine (arrow). In this example, the pipette solution contained 100 µM CNQX, and upon local injection of 1 µL, there was a reduction in the amplitude of LFP negativities. (**B**) Mean LFP changes obtained under control conditions (n=21) were reduced after the application of 100 µM CNQX (n=8), 100 µM AP5 (n=5), or both 100 µM CNQX and 100 µM AP5 (n=8). (**C**) Box plot illustrating how the initially recorded peak negativities were affected by the application of 100 µM CNQX or 100 µM AP5 together with 100 µM CNQX. Boxes represent the median (horizontal line), 25th to 75th quartiles, and ranges (whiskers) of the indicated experimental groups. Statistical differences were evaluated using a paired t-test. (**D**) Rubi-glutamate was injected

*Figure 2 continued on next page*

*Figure 2 continued*

into the GFP-labeled glomerulus and locally uncaged with a 500 ms pulse of blue light (circle). The example shows the change in LFP induced by two flashes (squares) delivered at an interval of 9 s. (E) Application of picrotoxin, a GABA$_A$ antagonist, did not modify odor-evoked changes. Recordings show average responses (n=9). Statistical differences were evaluated using a paired t-test. (F) Odor-evoked LFP changes were exclusively triggered by ipsilateral stimulation. Individual responses are indicated in gray, with the representative average response in black. Contralateral stimuli (yellow) did not modify the LFP, as shown in the average representative response (brown).

## Glomerular responses depend on the number of input neurons

The amplitude of LFP negativities decreases during the reinnervation of the glomerular layer after olfactory nerve transection (*Terni et al., 2017*), which suggests that LFP changes are related to the number of OSNs axons projecting to glomeruli. If odor-evoked responses were indeed related to glomerular input, the smallest negativities should take place in the earliest developmental stages. This possibility was confirmed by observing that GFP-positive neurons found in the olfactory epithelium increased with normal development, and their number followed an empirical linear relationship related to olfactory nerve width for tadpoles found between NF stages 47 and 54 (*Figure 3A*, $r^2$=0.96). The peak of the LFP response also varied with respect to olfactory nerve width but followed an exponential fit. Negativities reached a steady-state value (*Figure 3A*), thus suggesting the achievement of a consolidated glomerular activation in this developmental time interval. The presence of constant glomerular responses for NF stages >50 coincides with the consolidation of the basic structure of the olfactory bulb, which remains constant throughout larval life (*Byrd and Burd, 1991*). LFP changes were empirically described by the function $LFP_{peak}=LFP_{steady-state}+Ae^{-r \cdot ONW}$ that associated the maximum change in the LFP (in µV) to olfactory nerve width (ONW, expressed in µm). The fit revealed an increase rate (r) of 0.13 µm$^{-1}$ starting at a diameter of 21 µm, which corresponds to the thinnest olfactory nerves present during development around NF stage 40. Considering the linear correlation found between the number of GFP-positive OSNs and the width of olfactory nerves, it was possible to rewrite the exponential fit as a function of the number of OSNs. The estimated peak negativity in the LFP could thus be obtained according to $LFP_{peak} = 108-1866 \cdot e^{-0.11* \cdot OSNs}$ for a given number of OSNs in the olfactory epithelium. On average, an olfactory epithelium contained 30±2 GFP-positive OSNs, ranging from 24 to 39 labeled neurons (n=44), and this relationship provided a quantitative association of the amplitude of odor-evoked responses to the insertion of OSNs in a glomerulus during normal development.

## Glomerular responses are potentiated by the injury of the contralateral olfactory pathway

Transection of both olfactory nerves causes a silencing of the capacity of *X. tropicalis* tadpoles to perceive odors (*Terni et al., 2017*). Since animals having a single nerve still display normal olfactory-guided behavior, we investigated how information is transmitted in the absence of the mirror pathway. Olfactory nerve transection caused a potentiation of evoked LFP negativities in the recorded contralateral olfactory glomerulus (*Figure 3B*) that was already observed 2 hr after injury (*Figure 3C*). All tadpoles investigated showed odor-evoked responses above the expected $LFP_{peak}$ level for the range of developmental stages analyzed. By reaching a maximum potentiation between 24 hr and 48 hr after transection, $LFP_{peak}$ responses returned to control values following an exponential recovery phase that took place with a time constant of 100 hr (*Figure 3D*). The overall duration of potentiation coincided with the approximately two weeks required to reform the glomerular layer upon olfactory nerve transection (*Terni et al., 2017*), suggesting that the increase in glomerular output occurred while the circuitry of the contralateral olfactory bulb was being remodeled after injury.

$LFP_{peak}$ responses had a strong synaptic contribution because they were dependent on the number of OSNs innervating the glomerulus (*Figure 3A*), were sensitive to the application of glutamate receptor blockers (*Figure 2A–C*), and could be triggered by glutamate uncaging (*Figure 2D*). These results complement those obtained in rats showing a correlation of the amplitude of LFP negativities to calcium influx in OSN terminals (*Lecoq et al., 2009*) and EPSPs recorded in mitral cells (*Chaigneau et al., 2007*). We thus hypothesized that the observed potentiation arises from glutamatergic synapses mediating glomerular activation.

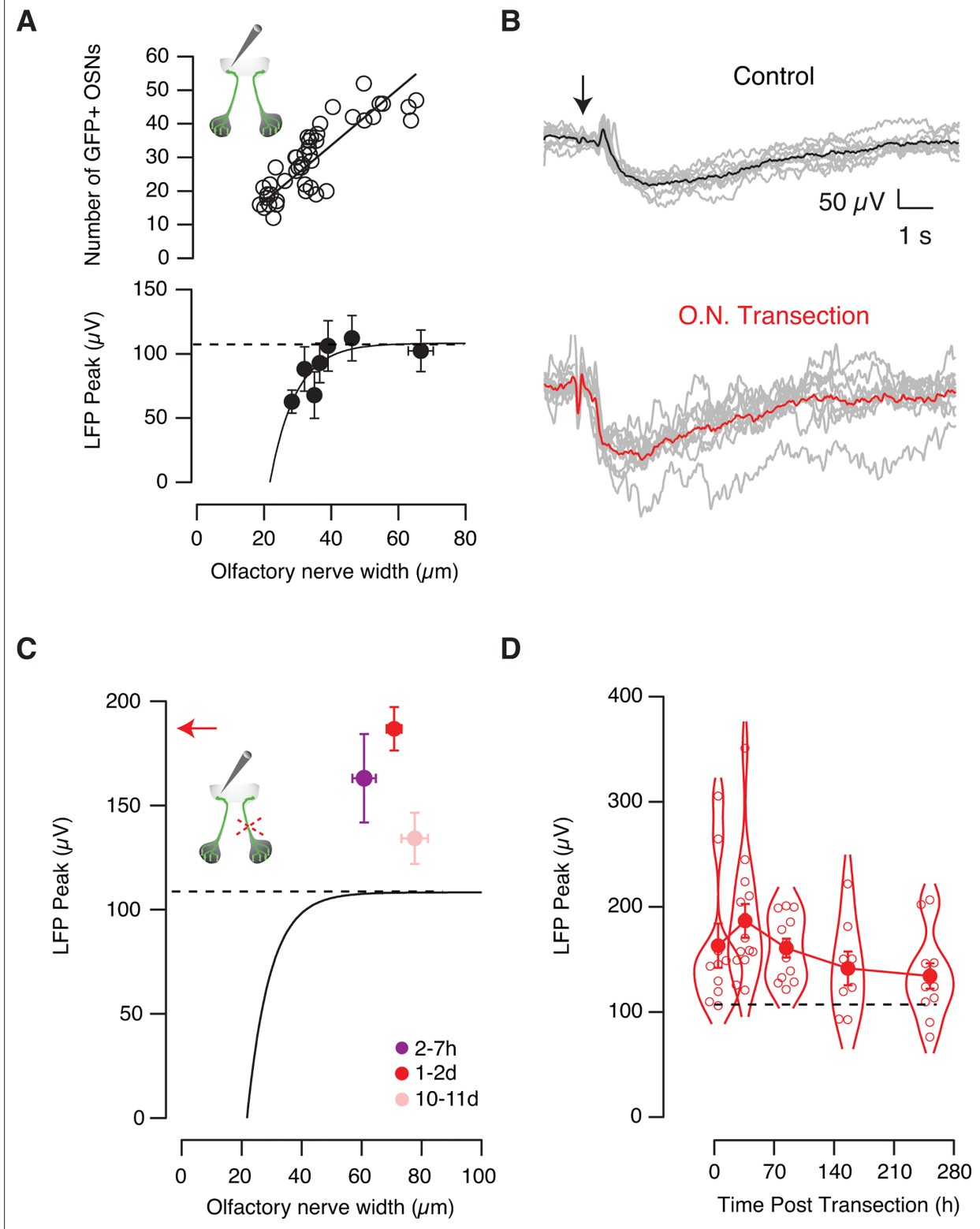

**Figure 3.** Potentiation of odor-evoked responses by transection of the contralateral olfactory nerve. (**A**) The number of GFP-positive olfactory sensory neurons (OSNs) and the amplitude of odor-evoked negative deflections of the local field potential (LFP) were related to olfactory nerve width according to linear and exponential functions, respectively. Individual data points are represented by circles (n=48). Each bin indicates the mean ± s.e.m. of n=6 tadpoles. The dotted line indicates the steady-state LFP amplitude reached during development. (**B**) Representative odor-evoked responses obtained in a control tadpole (black) and in a different animal (red), 24 hr after contralateral nerve transection. Gray traces indicate individual responses to the

*Figure 3 continued on next page*

*Figure 3 continued*

application of 200 µM methionine solution (arrow). (**C**) Odor-evoked LFP changes exhibit amplitudes above the expected values (dotted line as in A) after contralateral olfactory nerve transection at the indicated time points. The dots represent the mean ± s.e.m. obtained 2–7 hr (n=10), 1–2 days (n=14), and 10–11 days (n=11) post-injury. There was a 75% increase in animals recorded 1–2 days after transection of the contralateral olfactory nerve (red arrow) compared to control tadpoles (dotted line). (**D**) Dots (mean ± s.e.m.) connected by a line illustrate odor-evoked glomerular responses at the indicated times after injury. The superimposed violin plot displays individual data. Most LFP$_{peak}$ values are above the level expected for the developmental period studied (dotted line as in **A**).

## Tissue repair mechanisms are not involved in the potentiation of odor-evoked responses

Since inflammatory mediators can affect synaptic functions (*Wu et al., 2015*), we evaluated the involvement of molecules released by injury using two different strategies. First, to observe if sensory nerve damage enhanced LFP$_{peak}$ responses, we sectioned both optic nerves (*Figure 4A*). Even though optic and olfactory nerves show substantial functional and anatomical differences, the nature of the injury, as well as the distance to the recorded site, was comparable between optic and olfactory nerve transection. On these bases, we assumed that the effect generated by diffusible inflammatory mediators, or the capacity to mobilize cells mediating inflammation, should be comparable in both experimental conditions. LFP negativities recorded 24–48 hr after bilateral optic nerve transection showed an amplitude of 86±6 µV (n=17). This value was expected for normal development but was significantly lower (p=1.92·10$^{-5}$, unpaired t-test) than the amplitude of 186±16 µV (n=14) found in tadpoles subjected to transection of the contralateral olfactory nerve (*Figure 4B*).

Second, we investigated the release of reactive oxygen species (ROS), which are key to triggering tissue repair mechanisms in tadpoles (*Love et al., 2013*). The *Xenopus laevis* line HyPer-YFP displays a ubiquitous expression of the $H_2O_2$ sensor HyPerYFP that allows the in vivo detection of ROS created by wounds (*Love et al., 2013*; *Niethammer et al., 2009*). Two hours after unilateral olfactory nerve transection, there was an increase in $H_2O_2$ levels that was restricted to the cells present at the injury site (*Figure 4C*). ROS levels remained unaltered in the olfactory epithelium and in both olfactory bulbs. Moreover, the amplitude of odor-evoked changes in the LFP was unaffected by the presence of 200 µM apocynin or 2 µM diphenyleneiodonium, two blockers of ROS production (*Figure 4D*, p=0.36, ANOVA followed by Tukey's test) that prevent tail regeneration at these concentrations (*Love et al., 2013*). These findings together suggested that the enhancement of glomerular responses was unlikely caused by inflammatory mediators or the activation of fundamental tissue repair mechanisms present in tadpoles and revealed the presence of an intrinsic compensatory mechanism operating in the olfactory system.

## Potentiation of glomerular responses is of presynaptic origin

The activation of individual glomeruli was imaged in the transgenic *X. tropicalis* line *tubb2b:GCaMP6s* to gain insight into a putative synaptic mechanism acting on the bilateral balance of odor-evoked responses. Ipsilateral application of 200 µM methionine to the olfactory epithelium activated a defined set of glomeruli, which were identified as round, 10–20 µm diameter structures. The same glomeruli reacted to three or more consecutive methionine stimulations (*Figure 5A*). Responses detected as calcium transients showed comparable amplitudes and temporal profiles (*Figure 5B*), suggesting that the simultaneous activation of sparse glomeruli contributed to the elaboration of an olfactory map. Two lines of evidence support that calcium increases originated in presynaptic terminals of OSNs. First, because the *tubb2b* promoter primarily targets OSNs over olfactory bulb interneurons. A detailed characterization carried out in *X. laevis* tadpoles shows that *tubb2b* efficiently drives expression of a genetically encoded fluorescent reporter in OSNs but fails to label periglomerular neurons positive for calretinin or tyrosine hydroxylase and only targets 24% of mitral cells (*Daume et al., 2022*). The convergence of multiple fluorescent OSN axon terminals on a single glomerulus, along with a minimal presence of labeled juxtaglomerular and mitral cells, makes *X. tropicalis tubb2b:GCaMP6s* tadpoles well suited to detect the presynaptic component of glomerular responses. Second, because the simultaneous imaging of glomerular activation and recording of LFP negativities revealed a temporal correlation of the two signals (*Figure 5C*). This observation matches the findings of a previous study carried out in rats, where a comparable relationship was described between responses detected in OSN axon terminals labeled with a high-affinity calcium dye and LFP signals (*Lecoq et al., 2009*). Although

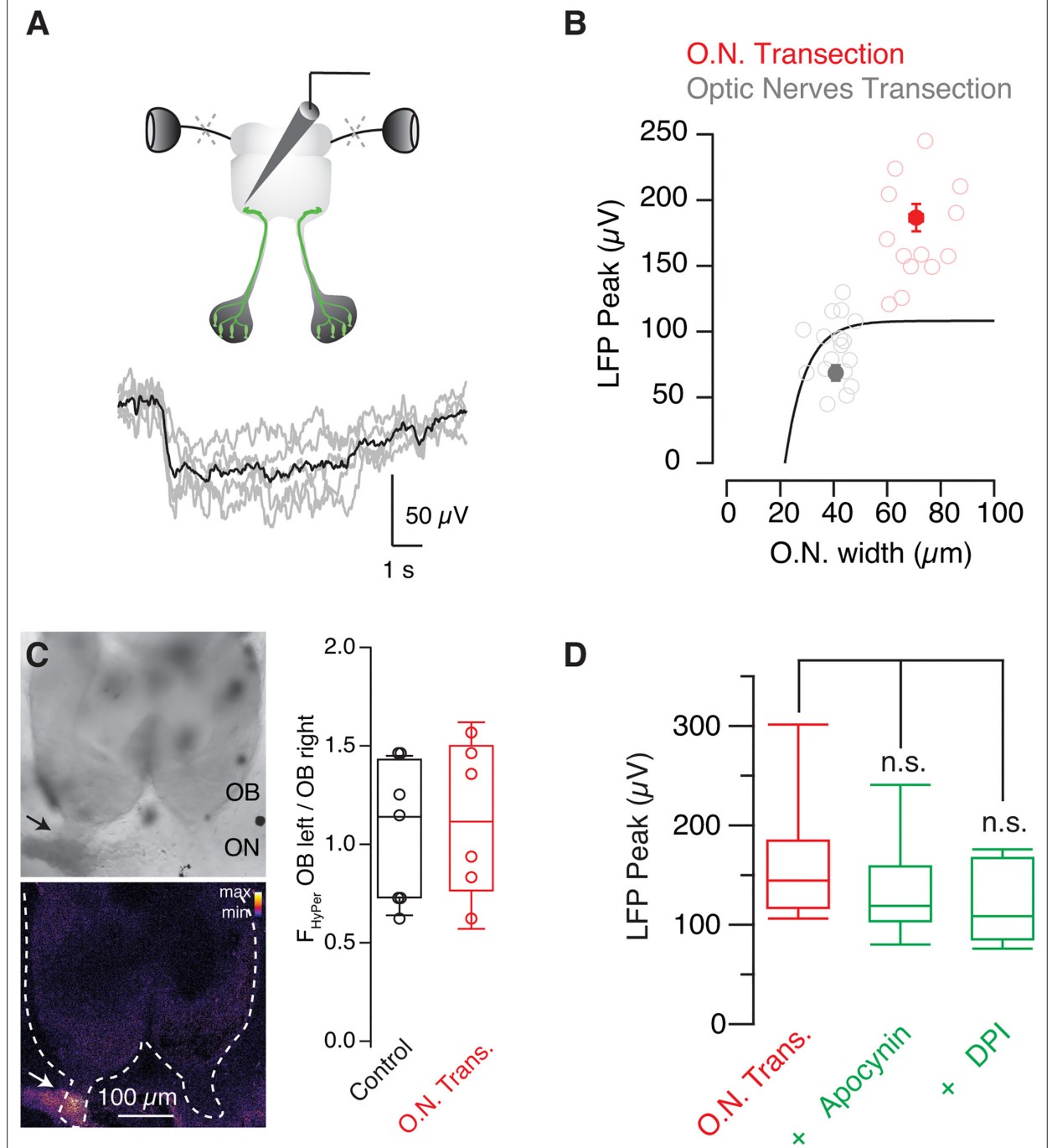

**Figure 4.** The potentiation of odor-evoked responses is not mediated by injury-derived cues. (**A**) Odor-evoked local field potential (LFP) changes were recorded by an electrode targeted to the GFP-positive glomerulus of *Dre.mxn1:GFP* tadpoles 1–2 days after bilateral transection of optic nerves. (**B**) Peak LFP negativities recorded in tadpoles with sectioned optic nerves (n=18) did not exhibit the characteristic potentiation observed after transection of the contralateral olfactory nerve (n=14), as they remained within the range of values observed during normal development (solid line, as in *Figure 3A*). Bins indicate mean ± s.e.m., circles show individual values. (**C**) Imaging of reactive oxygen species (ROS) 2 hr after transecting one olfactory nerve (arrow). The ratio between the fluorescence emitted by HyPer-YFP when excited at 488 nm and 405 nm is indicated in pseudocolor. Notice that ROS were increased at the injury site but remained at basal levels in both olfactory bulbs, as indicated by the box plot. Each circle shows values collected from a single tadpole. (**D**) Blocking of ROS production by incubating tadpoles with 200 µM apocynin (n=10) or 2 µM diphenyleneiodonium (DPI, n=5) did not modify the amplitude of odor-evoked LFP responses recorded 24 hr after contralateral olfactory nerve transection (n=10). Boxes represent the median (horizontal line), 25th to 75th quartiles, and ranges (whiskers) of the indicated experimental groups.

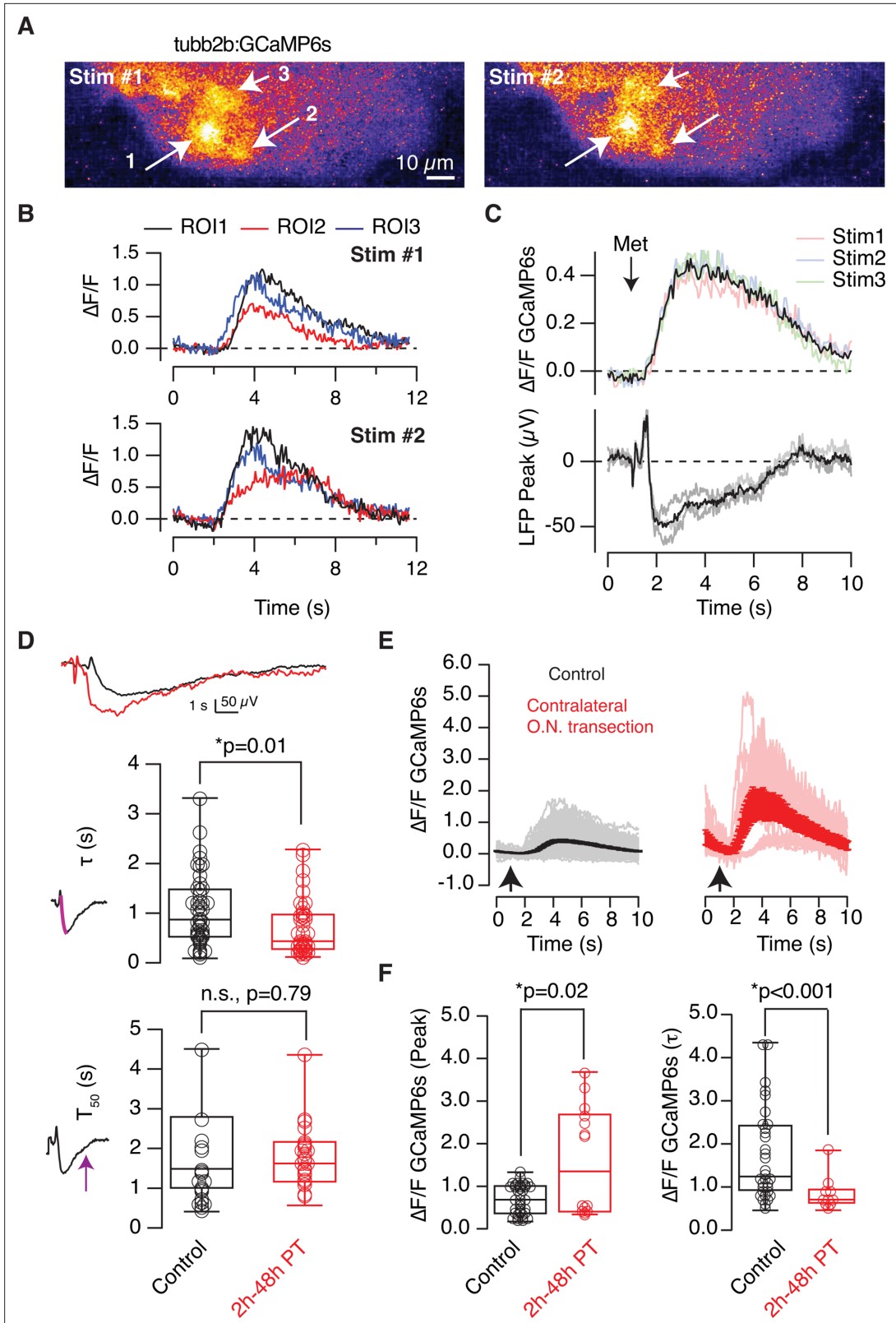

**Figure 5.** The presynaptic component of glomerular activation is affected by damage to contralateral olfactory sensory neurons. (**A**) Application of a puff of 200 µM methionine to the olfactory epithelium activates a set of glomeruli in the ipsilateral olfactory bulb (arrows) of *tubb2b:GCaMP6*s tadpoles. Images show the relative changes in GCaMP6s fluorescence (ΔF/F) obtained after two sequential stimulations carried out in a single tadpole. (**B**) Time course of the responses detected in the glomeruli indicated in (**A**). (**C**) An example showing the simultaneous recording of local field potential (LFP) and

*Figure 5 continued on next page*

*Figure 5 continued*

changes in GCaMP6s fluorescence in the region targeted by the electrode. Colored traces and gray traces show the change in GCaMP6s fluorescence (ΔF/F) and LFP, respectively, observed after three sequential applications of 200 μM methionine. Black traces show the average ΔF/F and LFP responses. (**D**) Kinetics of the change in LFP observed in tadpoles with the contralateral olfactory nerve transected between 2 hr and 48 hr prior to recording. The differences are illustrated by representative recordings obtained in two different tadpoles. (**E**) Intracellular calcium increases detected in glomeruli of control tadpoles with intact olfactory pathways (35 glomeruli, 10 tadpoles, black), and, in tadpoles subjected to the transection of the contralateral olfactory nerve (10 glomeruli, 3 tadpoles, red). Each trace indicates the response of a glomerulus to a single stimulus. Solid lines and error bars indicate mean ± s.e.m. (**F**) Calcium transients detected in tadpoles with an olfactory nerve transected showed a larger amplitude and a rising phase with a shorter time constant ( τ ). Boxes in (**D**) and (**F**) represent the median (horizontal line), 25th to 75th quartiles, and ranges (whiskers) of the indicated experimental groups. Statistical differences in (**D**) and (**F**) were evaluated using paired and unpaired t-tests, respectively. Circles in (**D**) indicate tadpoles and in (**F**) refer to glomeruli.

tadpoles of the *tubb2b:GCaMP6s* line consistently showed glomeruli activated by methionine, the coupling of imaging to LFP signals had a low success rate. Even though the electrode was targeted to the lateral glomerular cluster, which is the region expected to concentrate responses to amino acids (*Weiss et al., 2021a*), LFP negativities were detected in only 33% of animals tested (n=17/52). This figure is below the 50% found after performing recordings in random locations of the glomerular layer (*Figure 1D*), but it is higher than the ~25% success rate found in *X. laevis* tadpoles (*Manzini et al., 2007*). This observation reinforces the selectivity of LFP recordings to detect the output of a single glomerulus, which in this set of experiments was coupled to the fluorescent response found in closest apposition to the tip of the pipette. The readouts of glomerular input and output were thus reported by transient increases in GCaMP6s fluorescence and LFP signals, respectively.

A hallmark of contralateral olfactory nerve transection was the development of larger LFP negativities with a faster onset. The time constant to reach the $LFP_{peak}$ shortened significantly (p=0.017, unpaired t-test) from 1.05±0.1 s (n=46 tadpoles) to 0.71±0.1 s (n=39 tadpoles), while the recovery phase was unaffected (*Figure 5D*, p=0.79, unpaired t-test). Changes in glomerular output likely had a presynaptic origin, because calcium transients were larger and showed faster onset kinetics after contralateral nerve transection (*Figure 5E and F*). The average amplitude increased more than twofold (p=0.02, unpaired t-test) and the time constant describing the rise time shortened significantly (p=0.0003, unpaired t-test) in tadpoles with a transected olfactory nerve compared to control animals (*Figure 5F*). The slow kinetics of GCaMP6s prevented a precise temporal association between calcium buildup and the onset of LFP changes (*Figure 5C*) and a certain saturation of the high-affinity calcium indicator in axon terminals was also expected. However, despite these limitations to quantitative analysis, the changes reported by GCaMP6s showed an overall enhancement of cytosolic calcium levels mediating neurotransmitter release. We next sought to investigate which regulatory mechanisms acting on OSN axon terminals were under bilateral control to alter glomerular output.

## Presynaptic inhibition driven by dopamine D$_2$ receptors is affected by contralateral glomerular input

Presynaptic inhibition regulates neurotransmission in olfactory glomeruli (*McGann, 2013*). In mice, discrete populations of juxtaglomerular neurons that release GABA or dopamine activate GABA$_B$ or D$_2$ receptors present in presynaptic terminals of OSNs to lower glutamate release (*Wachowiak et al., 2005*; *Wachowiak and Cohen, 1999*; *Ennis et al., 2001*). Evidence indicates that this is a tonically active mechanism (*Pírez and Wachowiak, 2008*), thus meaning the normal processing of odor information relies on a certain level of inhibition of glomerular output. Since several previous works identified neurons immunoreactive for tyrosine hydroxylase or GABA innervating the glomerular layer of *X. laevis* tadpoles (*González et al., 1994*; *Daume et al., 2022*; *Nezlin and Schild, 2000*), dopamine and/or GABA could also be mediating glomerular inhibition in *X. tropicalis* tadpoles.

We identified TH-positive neurons (TH+) at the border of the glomerular and the mitral cell layers (*Figure 6A*). The morphological characteristics of these neurons were reminiscent of type-1 TH neurons described in the olfactory bulb of adult frogs, associated with a dopaminergic phenotype (*Boyd and Delaney, 2002*). The processes of TH+ neurons ramified within the glomerular layer, suggesting the establishment of relationships to several glomeruli (*Figure 6B*). The lateral glomerulus formed by GFP-positive axon terminals was contacted by TH+ puncta, thus indicating its function was modulated by dopamine signaling (*Figure 6C*).

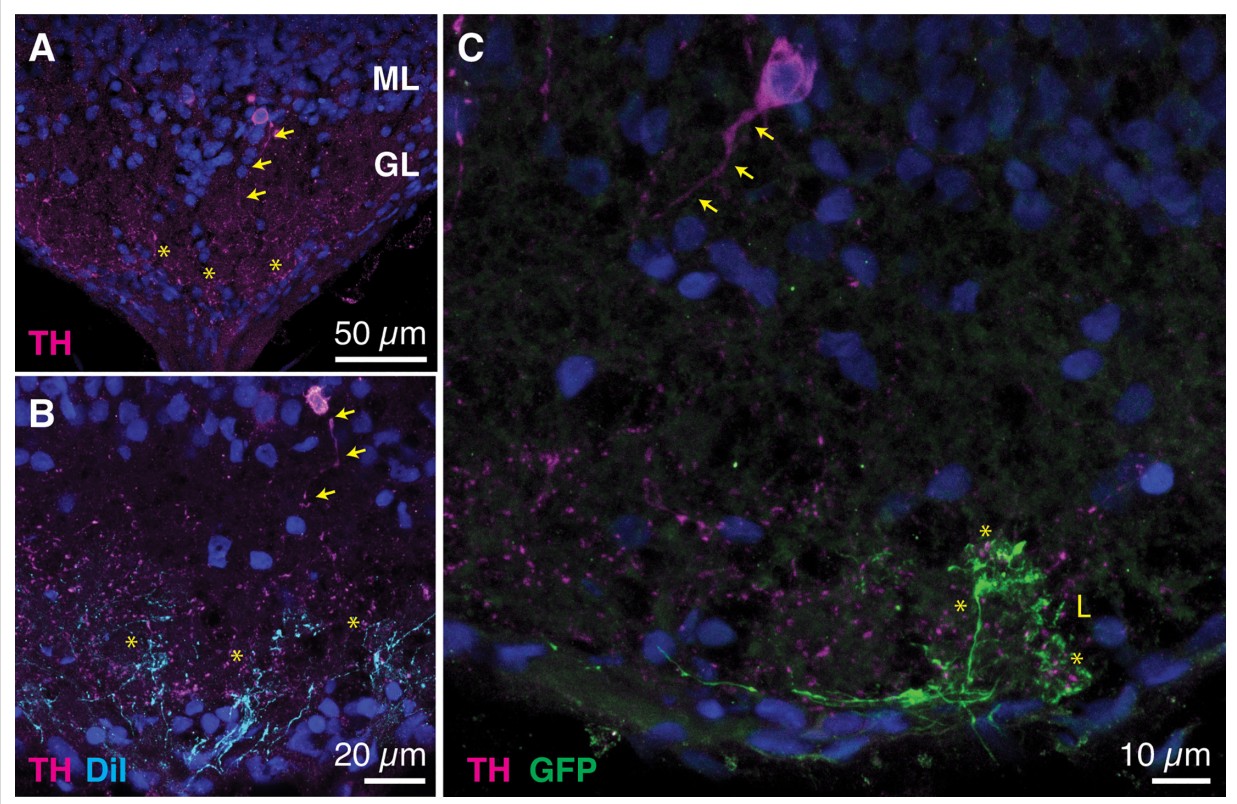

**Figure 6.** Tyrosine hydroxylase positive neurons project to the glomerular layer of the olfactory bulb. (**A**) Cell bodies of neurons expressing tyrosine hydroxylase (TH+, magenta) were sparsely distributed at the border of the glomerular (GL) and mitral cell (ML) layers of the olfactory bulb and sent their neuronal processes (arrows) to innervate the glomerular layer (asterisks). (**B**) Projections of TH+ neurons (arrows) contacted axon terminals of olfactory sensory neurons labeled with DiI (cyan, asterisks). (**C**) The lateral glomerulus (**L**) labeled in Dre.mxn1:GFP *X. tropicalis* tadpoles was contacted by projections (asterisks) of processes emerging from TH+ neurons (arrows).

Odor-evoked responses were enhanced by raclopride, a $D_2$ antagonist, in tadpoles with intact olfactory pathways (*Figure 7A*). The average negativity of 73±5 µV (n=8) experienced a gradual increase in the presence of 300 nM raclopride that reached 128±16 µV, 20 min after its application (*Figure 7B*, p=0.012, paired t-test). The potentiation of odor-evoked responses caused by the $D_2$ antagonist was not observed in tadpoles with the contralateral olfactory nerve transected. Baseline negativities of 133±24 µV (n=11) were minimally increased by raclopride, changing to 156±33 µV (n=11, p=0.16, paired t-test), 20 min after drug application.

The time course of the increase in $LFP_{peaks}$ observed in control tadpoles was well described by a Hill equation, reporting a time required to reach the 50% of the maximal increase of approximately 20 min (*Figure 7C*). The time window describing the effect of raclopride could be attributed to disinhibition, since the primary target of presynaptic $D_2$ receptor activation is the decrease of cAMP levels and a consequent lowering of neurotransmitter release (*Kaneko and Takahashi, 2004*). The potentiation mediated by raclopride was comparable to the increase in $LFP_{peaks}$ caused by olfactory nerve transection (*Figure 7B*, p=0.46, unpaired t-test). These results indicated that the tonic inhibition of glomerular output was comparably reduced either by directly inhibiting $D_2$ receptors or by surgically transecting the contralateral olfactory nerve. Further evidence for the involvement of $D_2$ receptors was obtained by simultaneously imaging GCaMP6s fluorescence and LFP signals (*Figure 7D*). In the illustrated example, raclopride shortened the time constant ($\tau$) to reach the calcium peak from 1.3 s to 0.5 s and potentiated LFP changes by 50%, thus reproducing the effect of contralateral olfactory nerve transection on glomerular input and output, respectively (*Figures 3C and 5F*). On average, raclopride reduced the onset time constant of calcium transients from 1.27 s to 0.8 s (n=4 tadpoles, p=0.02, paired t-test), evidencing that the potentiation of glomerular responses was mediated by a

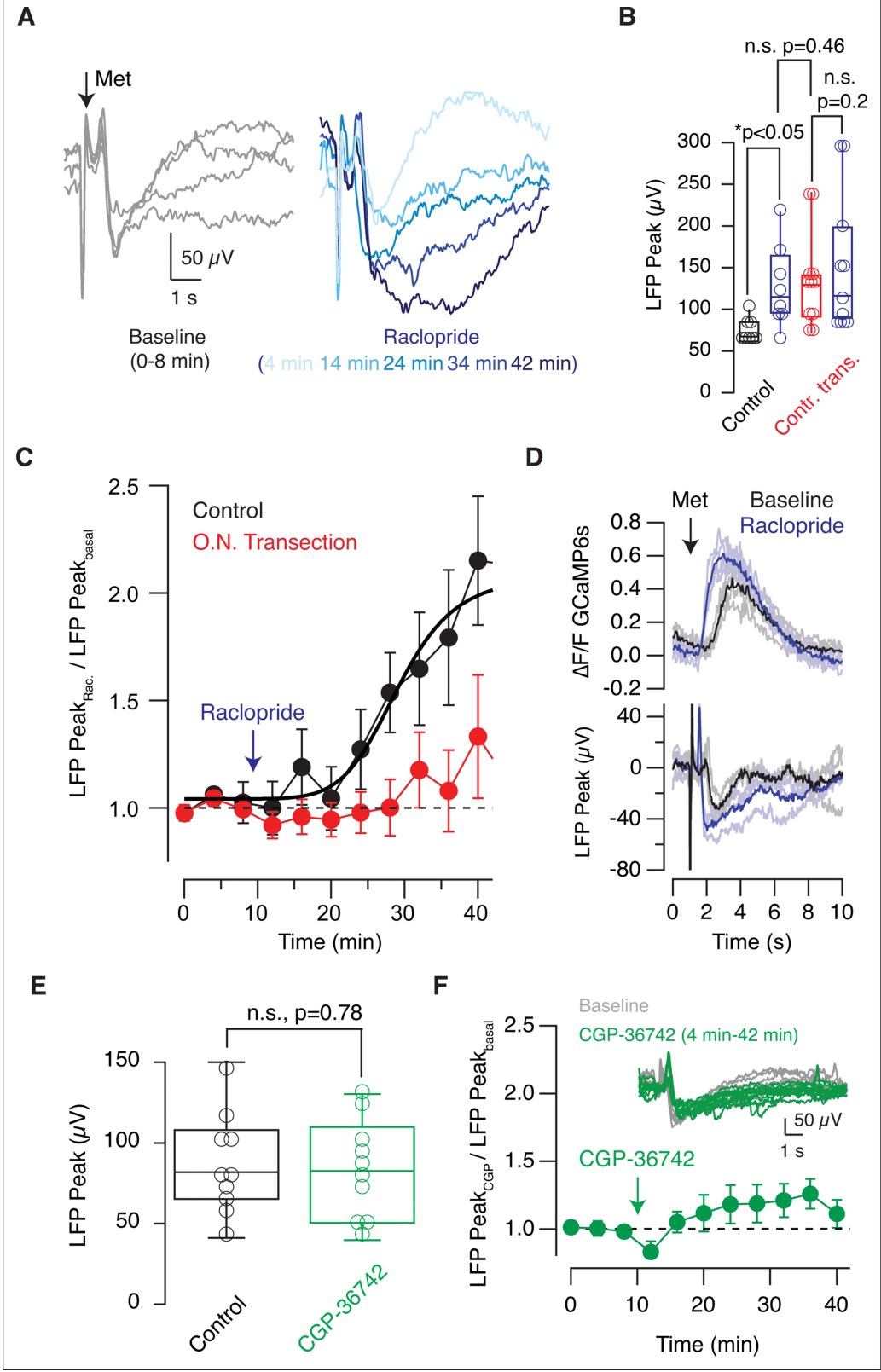

**Figure 7.** Contralateral input modulates presynaptic inhibition mediated by dopamine $D_2$ receptors and is involved in the potentiation of glomerular responses. (**A**) Recordings obtained in a control tadpole showing how the amplitude of local field potential (LFP) responses (gray traces) obtained during a baseline period of 8 min increased in a time-dependent manner after local application of 300 nM raclopride, a $D_2$ receptor antagonist.

*Figure 7 continued on next page*

*Figure 7 continued*

(**B**) Box plot showing the effect of 300 nM raclopride (blue) on the amplitude of LFP responses recorded in tadpoles with full capacity to process odors (control, n=8, black) and tadpoles subjected to the transection of the contralateral olfactory nerve (n=11, red). Boxes represent the median (horizontal line), 25th to 75th quartiles, and ranges (whiskers) of the indicated experimental groups. The effect of raclopride was evaluated using the paired t-test, and the comparison between control and transected groups was performed using the unpaired t-test. (**C**) Relative change in LFP responses induced by 300 nM raclopride in control tadpoles and in tadpoles subjected to the transection of the contralateral olfactory nerve. Dots represent mean ± s.e.m. The solid black line illustrates the fit to a Hill equation, defining a $T_{50}$ at 20 min. (**D**) Simultaneous recording of LFP and changes in GCaMP6s fluorescence in a *tubb2b:GCaMP6s* tadpole. Gray traces and light blue traces show individual responses to sequential stimulations before and after application of 300 nM raclopride, respectively. Average responses are shown in black and dark blue. (**E**) Application of CGP-36742, a GABA_B receptor antagonist, did not modify LFP responses. The box plot compares the amplitude of LFP changes recorded before (gray) and 20 min after local application of 300 µM CGP-36742 (green). Statistical differences were evaluated using paired t-test. (**F**) Time course of relative LFP changes induced by 300 µM CGP-36742. Dots represent mean ± s.e.m. (n=10). The inset shows recordings obtained in a tadpole in baseline conditions (gray) and after injection of CGP-36742 (green).

faster buildup of presynaptic calcium levels that took place after antagonizing the inhibitory action of $D_2$ receptors.

To explore a concomitant inhibition mediated by presynaptic GABA_B receptors (*McGann, 2013*), we investigated the effect of the selective GABA_B antagonist CGP-36742. LFP_peaks remained unchanged (*Figure 7E and F*, p=0.77, paired t-test). A significant inhibition by GABA was thus ruled out, suggesting that dopamine was the main neurotransmitter mediating tonic inhibition of glomerular output in *Xenopus* tadpoles.

## Effect on glomerular responses of the partial elimination of mirror OSNs

To shed light on the involvement of chemotopy in the contralaterally driven potentiation of glomerular output, we modified the two-photon chemical apoptotic targeted ablation (2Phatal) technique (*Hill et al., 2017*) to eliminate groups of GFP-positive OSNs using a conventional confocal microscope (*Figure 8A*). Tadpoles were placed for 15 min in *Xenopus* water containing 5 µg/mL Hoechst 33342, which is a membrane-permeable dye with affinity for nucleic acids, to label all cells found within the olfactory epithelium. Damage was exerted by photobleaching groups of cells found in regions of interest (ROIs) that contained two or more GFP-positive OSNs. Efficient photobleaching of the nuclear label caused cell death, which was confirmed by the observation of condensed nuclei after 24 hr (*Figure 8A*). The use of the confocal microscope precluded the achievement of single-cell ablation but supported the robust elimination of cells within ROIs. Olfactory epithelial cells found outside the selected areas remained unaffected. Photobleaching was typically carried out in 4–6 ROIs, thus resulting in the elimination of 10–15 OSNs. Considering an epithelium contained on average ~30 GFP-positive OSNs, the protocol led to an estimated 30–50% reduction of OSNs innervating the right GFP-labeled glomerulus. Since the viability of tadpoles subjected to a more extensive reduction of fluorescent OSNs was compromised, all animals investigated still had a significant number of functional GFP-labeled OSNs after photoablation.

A characteristic of the recorded odor-evoked glomerular responses was their reproducibility. For a given tadpole, LFP negativities occurred after successive stimulations with a comparable profile (*Figures 1B, 2A, and 4A*) and showed an average variance in their amplitude of $375\pm92$ µV² (n=17). The variance of the responses increased more than threefold in tadpoles subjected to transection of the olfactory nerve (see, for example, *Figure 3B*), reaching an average value of $1488\pm280$ µV² (n=14). An enhancement of variance was thus associated with the potentiation observed in response to olfactory nerve transection. Incubation of tadpoles with Hoechst 33342 neither modified the amplitude nor the variance of LFP_peaks (*Figure 8B*). LFP negativities in tadpoles presenting a decimated population of contralateral GFP-positive OSNs showed an amplitude of $78\pm7$ µV (n=14) and a variance of $771\pm351$ µV², both being comparable to controls (*Figure 8C*). These values were also similar to tadpoles where regions lacking GFP-positive OSNs were photobleached.

These results suggest that potentiation driven by damage of contralateral OSNs was unrelated to chemotopy, which could be consistent with the innervation of several glomeruli by a single TH+

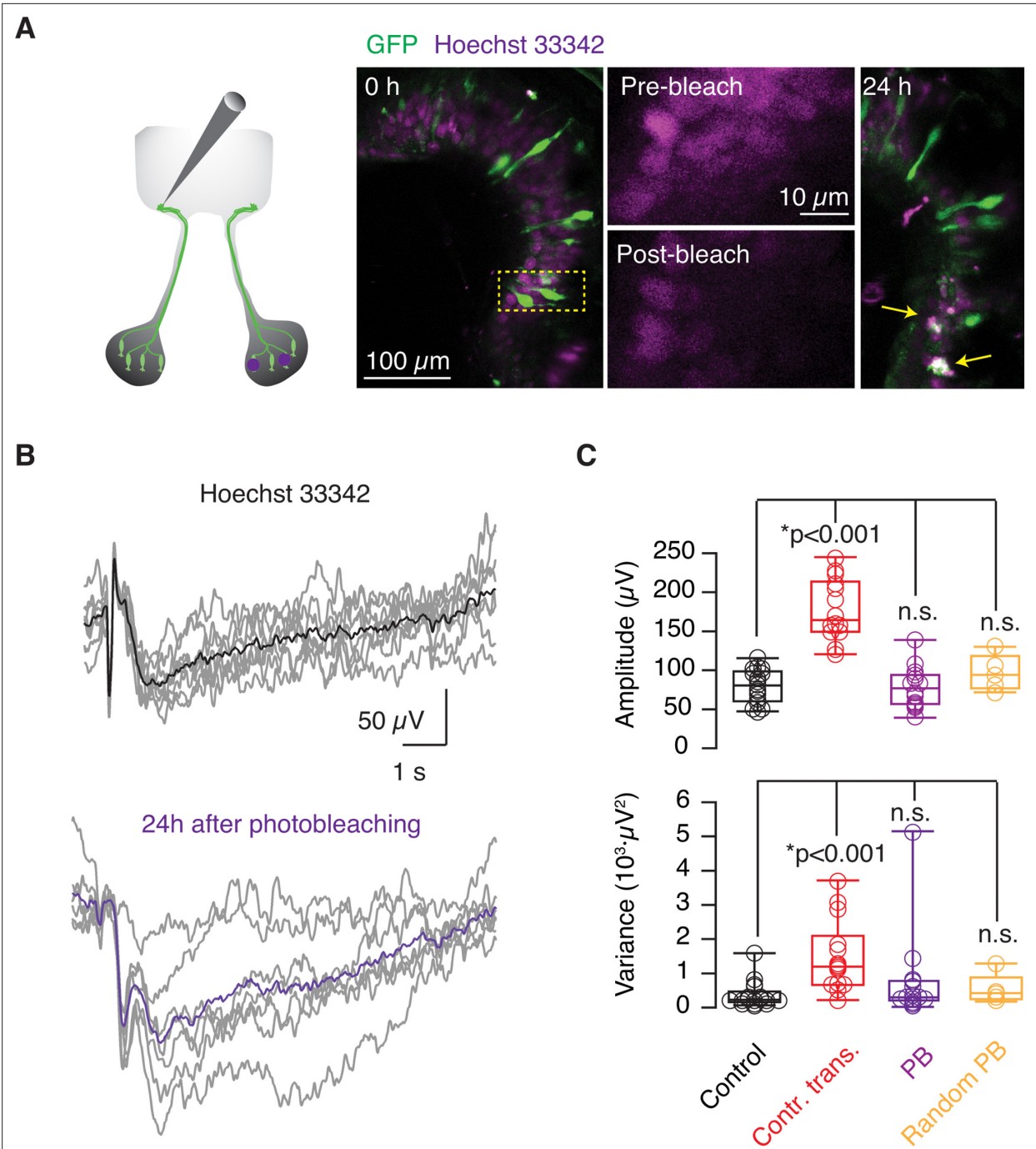

**Figure 8.** Effect on odor-evoked responses of selective photoablation of olfactory sensory neurons (OSNs) innervating the homologous contralateral glomerulus. (**A**) Odor-evoked local field potential (LFP) changes were recorded 1 day after the selective elimination of OSNs located in the right nasal cavity. After the identification of GFP-positive OSNs in the epithelium labeled with the nuclear marker Hoechst 33342, regions containing ≥2 fluorescent neurons were identified and photobleached. Cell targeting was confirmed by the suppression of the nuclear label. Only cells found within the photobleached areas exhibited fragmented nuclei (arrows) 24 hr after photobleaching. (**B**) Examples showing odor-evoked responses recorded in a tadpole incubated with Hoechst 33342 (black, average) and in a different tadpole 24 hr after the photobleaching of selected regions in the contralateral olfactory epithelium (violet, average). (**C**) Photoablation of GFP-positive neurons did not modify the amplitude or variance of contralateral odor-evoked responses recorded in the glomerulus innervated by cognate neurons. Boxes represent the median (horizontal line), 25th to 75th quartiles, and ranges (whiskers) of the indicated experimental groups. Statistical differences were evaluated using ANOVA followed by Tukey's test. Circles indicate values obtained from single tadpoles.

neuron (*Boyd and Delaney, 2002*). However, a reduction between 1/3 and 1/2 of mirror contralateral input might be insufficient to drive a significant increase in LFP$_{peaks}$ amplitude or variance, because only the complete ablation of GFP-labeled neurons could replicate the effect of transection. Consequently, it was not possible to rule out that the gain of function caused by contralateral injury was related to a graded contribution of each OSN to the total input of topographically related glomeruli.

## Pallial neurons are involved in the potentiation of glomerular responses induced by contralateral injury

To shed light on the circuit mediating the contralaterally driven potentiation of glomerular responses, we considered three theoretical scenarios. First, a direct bilateral innervation of glomerular layers by OSNs. Second, an interhemispheric association of glomeruli mediated by olfactory bulb projection neurons. Third, the involvement of the lateral pallium, which is considered to play a key role in olfactory processing in amphibians (*Moreno et al., 2008*; *Roth et al., 2007*).

The axons of OSNs labeled in the *Dre.mxn1:GFP X. tropicalis* formed the lateral glomerulus and showed some projections to medial glomerular structures (*Figure 1E*), but we did not find evidence for crossings at the level of the anterior commissure or midline. The possibility that GFP-labeled neurons displayed the properties of those innervating the γ-glomerulus (*Kludt et al., 2015*) was thus ruled out. In premetamorphic stages, most OSNs project to ipsilateral glomeruli (*Weiss et al., 2021b*) and, considering the lack of support for interhemispheric connections among olfactory bulb neurons as reported in zebrafish (*Kermen et al., 2020*), we thus evaluated the third possibility by investigating the capacity of pallial neurons to respond to odor stimulation.

ROIs measuring 30 µm in diameter were selected in the dorsolateral pallium of *tubb2b:GCaMP6*s tadpoles. All of them displayed synchronous, rhythmic calcium transients that occurred at frequencies ranging from 0.1 Hz to 0.5 Hz. Stimulation of the ipsilateral olfactory epithelium with 200 µM methionine evoked a response in all selected regions that was consistently observed during consecutive stimulations (*Figure 9A*). These observations confirmed the involvement of the dorsolateral pallium in the processing of olfactory information.

If pallial neurons were participating in the potentiation of contralateral glomerular output, their damage should induce a similar gain of function to that evoked by olfactory nerve transection (*Figure 3A*). Using iridectomy scissors, we made an injury at the level of the dorsal pallium in the medial to the lateral direction and recorded odor-evoked responses 1–2 days afterward (*Figure 9B*). An ~70% increase in odor-evoked responses was observed. The average amplitude of LFP$_{peaks}$ was 135±11 µV (n=25), significantly higher than responses found in control tadpoles (*Figure 9C*, p=6·10$^{-5}$, unpaired Student's t-test). These results showed that pallial neurons participated in the control of glomerular output and suggested that they could modulate the activity of contralateral dopaminergic neurons mediating presynaptic inhibition of OSN axon terminals.

## Discussion

Odor-evoked responses were recorded as changes of the LFP in an olfactory glomerulus labeled in the *Dre.mxn1:GFP X. tropicalis* line (*Figure 1*) and showed the following characteristics: (i) were mediated by glutamatergic synapses (*Figure 2A–C*), (ii) were triggered by ipsilateral stimulation of OSNs (*Figure 2F*), and (iii) exhibited an amplitude related to the number of input OSNs (*Figure 3A*). These characteristics are comparable to responses recorded in the glomerular layer of living rats where negativities are supported by the activation of glutamate receptors and locked to the respiration frequency (*Lecoq et al., 2009*). Glomerular responses were potentiated by the complete silencing of OSNs projecting to the contralateral olfactory bulb, thus demonstrating that the processing of information in the olfactory glomeruli of *Xenopus* tadpoles is not exclusively unilateral and is shaped contralaterally. The observed potentiation was not related to inflammatory mediators associated with injury because it was caused by a release of the inhibition made by D$_2$ dopamine receptors present in OSN axon terminals.

The similarities between the recordings obtained (*Figure 1B*) and those described in rats (*Chaigneau et al., 2007*) indicate the presence of evolutionarily conserved morphological and functional features between both species and, therefore, the applicability of findings described here to vertebrates. Our results achieved using glutamate blockers and uncaging of Rubi-glutamate show that the onset of

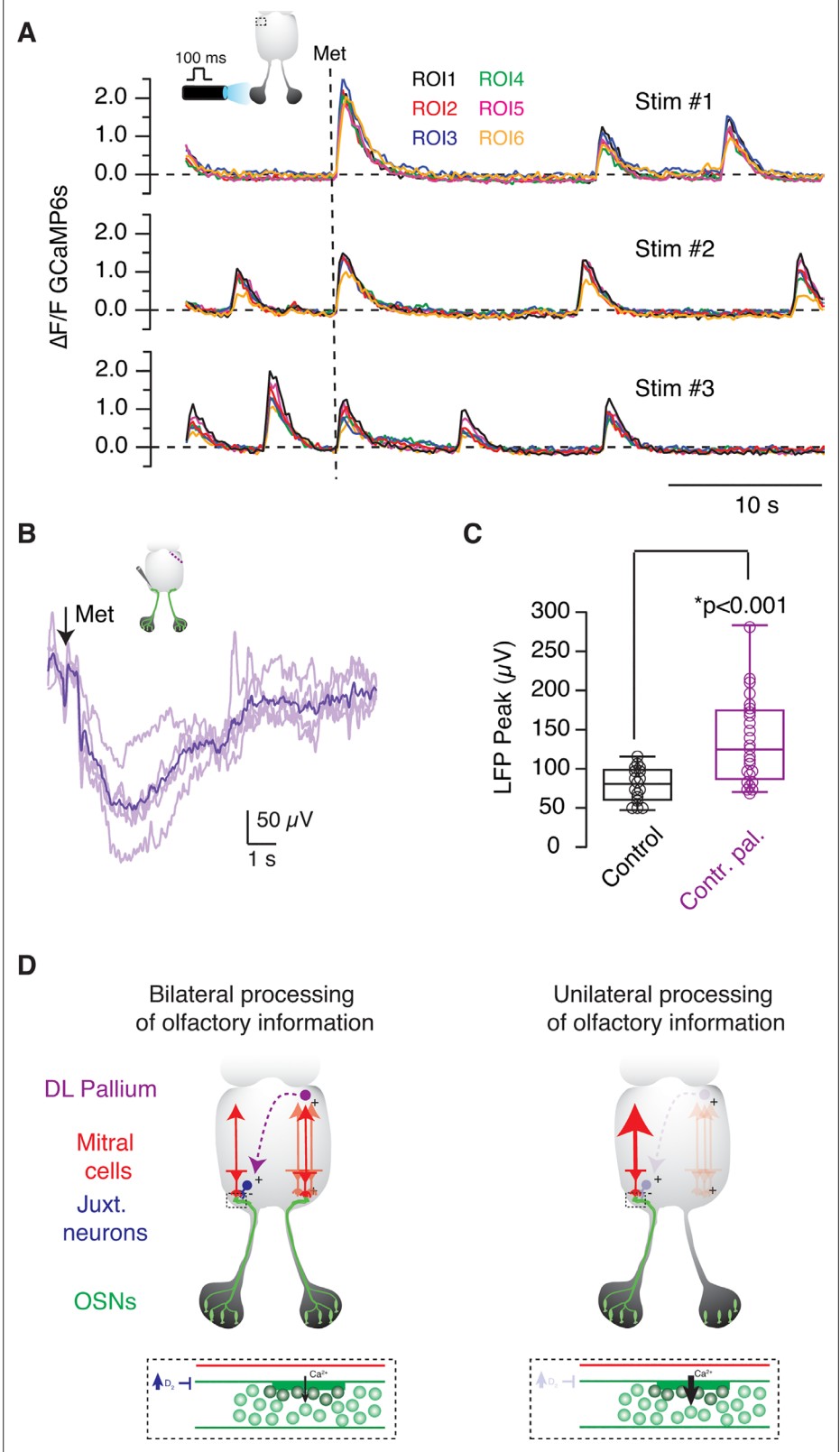

**Figure 9.** Pallial neurons are involved in the potentiation of glomerular responses driven by contralateral injury.
(**A**) Rhythmic calcium transients were detected in six different regions of interest (ROIs) located in the dorsolateral
pallium of a *X. tropicalis tubb2b:GCaMP6*s tadpole. Three consecutive stimulations carried out by applying
200 µM methionine to the ipsilateral olfactory epithelium (dotted line) evoked a synchronous response in the

*Figure 9 continued on next page*

*Figure 9 continued*

ROIs investigated. (**B, C**) Odor-evoked responses were recorded 24–48 hr after making a tangential injury in the contralateral dorsolateral pallium. The amplitude of LFP changes significantly increased in injured tadpoles. Boxes represent the median (horizontal line), 25th to 75th quartiles, and ranges (whiskers) of the indicated experimental groups. Statistical differences were evaluated using an unpaired t-test. (**D**) Model proposed for the bilateral modulation of glomerular output in the olfactory bulb of *Xenopus* tadpoles. A population of juxtaglomerular neurons releases dopamine to inhibit glomerular output by activating presynaptic D2 receptors present in olfactory sensory neurons (OSNs) (dotted box). The constant presence of dopamine within glomeruli is favored by the activity of the contralateral olfactory bulb. When the contribution of the contralateral pathway is suppressed, dopamine release diminishes, and glomerular responses become potentiated. The contralateral modulation of the tonic activity of dopaminergic juxtaglomerular neurons corrects for input differences and equalizes the synaptic output of olfactory glomeruli to achieve a bilaterally balanced transfer of information. The activity of dopaminergic interneurons is likely controlled by pallial neurons through a yet undetermined connectivity, taking advantage of their participation in the processing of olfactory information.

LFP changes detected in glomeruli is determined by glutamate release from OSNs. The ionic bases supporting the recovery phase of negativities are uncertain but, considering that responses obtained by glutamate uncaging were locked to the period defined by the light pulse, the participation of sustained neurotransmitter release is conceivable. This view is also supported by imaging of OMP-synapto-pHluorin mice, where continuous synaptic vesicle exocytosis occurs over several seconds in OSN axon terminals (*Petzold et al., 2008*).

The mammalian and amphibian olfactory bulbs display comparable anatomical and cellular organization (*Manzini et al., 2022*); however, it is yet unknown whether the three classes of juxtaglomerular neurons present in rodents – periglomerular, short-axon, and external tufted – also exist in *Xenopus*. In mice, short-axon cells are the main source of dopamine to olfactory glomeruli (*Kiyokage et al., 2010*); however, there is a lack of evidence for an equivalent neuronal type in the olfactory bulb of *Xenopus*. Here, we show that the processes of TH+ neurons extensively innervate the glomerular layer, thus indicating a significant role for dopaminergic signaling, in agreement with morphological observations (*González et al., 1994*). Our results indicate that a major role of dopamine is mediating the presynaptic inhibition of OSN axon terminals through D2 receptors. In rodents, presynaptic inhibition of OSNs is mediated by GABA acting through GABA$_B$ receptors and dopamine (*McGann, 2013*), and both neurotransmitters are typically co-released by specific types of juxtaglomerular neurons (*Kosaka et al., 2020*). Since we did not find evidence for the participation of GABA$_B$ receptors in inhibiting glomerular responses, the data obtained support a segregation of dopaminergic and GABAergic pathways, in agreement with morphological evidence (*Boyd and Delaney, 2002*).

To understand how tonic dopamine release in the glomerular layer of *Xenopus* tadpoles is affected by the number of contralateral OSNs, it is necessary to have a comprehensive understanding of the different types of interneurons present in the olfactory bulb of *X. tropicalis* tadpoles, which is currently lacking. The dopaminergic neurons present in the border of the glomerular and mitral cell layers of *X. tropicalis* tadpoles send their projections to glomeruli, which fit with the characteristics of type-1 TH+ interneurons described in adult frogs (*Boyd and Delaney, 2002*). The dendrites of type-1 TH+ interneurons innervate glomeruli and when type-1 TH+ neurons contain an axon, it recurves and projects to the glomerular layer. In mice, there are five different types of morphologically distinct juxtaglomerular dopaminergic neurons (*Kosaka et al., 2020*), and those that are anaxonic release dopamine from their dendritic tree, while those that bear axons release it from their axon (*Dorrego-Rivas et al., 2025*). Considering that an axon was not observed in all type-1 TH+ interneurons (*Boyd and Delaney, 2002*), it is conceivable that axon-bearing and anaxonic dopaminergic neurons could also exist in *Xenopus* tadpoles and both innervate glomeruli. If the majority of type-1 TH+ were spontaneously active, as reported in mice for dopaminergic juxtaglomerular neurons (*Pignatelli et al., 2005*), dopamine would thus play a key modulatory role in glomerular neurotransmission in *X. tropicalis* tadpoles.

Our results support that dopamine release is affected by the number of operative contralateral OSNs so that, in the extreme situation where all mirror olfactory pathways are silenced, dopamine secretion decreases and most of the inhibition is released (*Figure 9D*). The consequence is an immediate increase in glomerular responses in the non-damaged pathway, resulting in a compensated output signal. The lateral pallium is a main center for olfactory processing in frogs (*Moreno et al., 2008*), and we provide evidence for its participation in the bilateral homeostasis of glomerular output.

The injury of contralateral pallial neurons evoked a potentiation of glomerular responses that reproduced the effect of olfactory nerve transection, which supports a regulation of dopamine release by the pallium. Considering the capacity of pallial neurons to establish ipsilateral and contralateral relationships (*Roth et al., 2007*; *Westhoff and Roth, 2002*), our results are consistent with the presence of a pallial connectivity controlling the firing of TH+ neurons that innervate the glomerular layer of the olfactory bulb.

The in vivo evidence presented shows that presynaptic $D_2$ receptors exert an inhibitory action on calcium buildup in OSN axon terminals. This finding is well aligned with previous observations obtained in mouse brain slices (*Ennis et al., 2001*) and describes a relevant biological scenario for the role of dopamine inhibition in the glomerular layer of the olfactory bulb. N-type voltage-gated calcium channels are a likely target of dopamine because they are present in presynaptic terminals in olfactory glomeruli (*Weiss et al., 2014*), and they can be inhibited through multiple mechanisms activated by $D_2$ receptors (*Kisilevsky and Zamponi, 2008*).

Altogether, our results illustrate a homeostatic mechanism used to compensate for a reduced contribution of contralateral OSNs via an enhancement of glomerular output. It is possible that the compensation observed upon transection of an olfactory nerve takes advantage of dopaminergic signaling pathways used for the encoding of olfactory information. The normal activity of presynaptic $D_2$ receptors present in OSN axon terminals could be contralaterally regulated through a homeostatic loop controlled by pallial neurons to correct for imbalances in glomerular input. Considering the evolutionarily conserved features of the *Xenopus* olfactory bulb, the mechanism described here could be broadly applicable to vertebrates.

# Materials and methods
## Animals
Ethical procedures were approved by the Institutional Animal Care and Use Committee of the regional government (Generalitat de Catalunya, experimental procedure #10753). *X. tropicalis* (RRID:NXR_1018) were housed and raised according to the standard protocols of the animal facilities of the University of Barcelona. Tadpoles were obtained by natural mating and kept in tanks at 25°C. *Xenopus* larvae are not sexually dimorphic in aquatic stages, and sex differences (male vs female) were not considered in the current study. *Xenopus* water conductivity was adjusted to 700 μs·cm⁻¹, pH = 7.5. Two- to three-week-old tadpoles, found at stages 47–52 of the Nieuwkoop-Faber criteria (*Nieuwkoop and Faber, 1956*), were used for the experiments. The animals of a defined experimental group came from >4 different natural matings. The assignment of tadpoles to a particular group was not explicitly randomized, and the range of 8–15 follows standard group size but not an explicit power calculation. Investigators were not blinded during the selection of animals into groups or during analysis.

The *X. tropicalis* transgenic line *Dre.mxn1:GFP* (RRID:NXR_1111) was used for the visualization of a discrete population of OSNs (*Terni et al., 2017*; *Terni and Llobet, 2021*) to carry out electrophysiological recordings in a genetically defined glomerulus. Labeling of OSNs was consistent in all animals of this transgenic line. Calcium imaging was performed in the *X. tropicalis* transgenic line *ElasGFP:Tubb2b-GCaMP6s* (RRID:NXR_1123). In vivo detection of ROS was carried out in tadpoles of the *X. laevis* transgenic line *Hsa.UBC-Gal4;UAS:HyPer-YFP* (RRID:NXR_0127).

For surgical procedures, *Xenopus* tadpoles were anesthetized in 0.02% MS-222 and transferred to a wet nitrocellulose filter paper placed under a stereomicroscope. Iridectomy scissors were used for the unilateral transection of an olfactory nerve or the bilateral transection of optic nerves. Tadpoles were returned to water tanks after surgery.

## Electrophysiology
*X. tropicalis* tadpoles were anesthetized in 0.02% MS-222, and the portion of skin covering the olfactory bulb was removed. Animals were transferred to a well fabricated in a dish coated with silicone elastomer. A coverslip restricted tadpole movements and left olfactory placodes and bulbs accessible (*Terni et al., 2018*). The dish was placed on the stage of an upright microscope (Zeiss, Axioexaminer A1, Oberkochen, Germany) and was continuously perfused with *Xenopus* Ringer containing (in mM): 100 NaCl, 2 KCl, 1 CaCl₂, 2 MgCl₂, 10 glucose, 10 HEPES, 240 mOsm/kg, pH = 7.8, supplemented with 1 μM d-tubocurarine to prevent muscle contractions. All salts and drugs were from Sigma-Aldrich

(Saint Louis, MO, USA). CGP-36742 was from Novartis Pharmaceuticals (Basel, Switzerland). To carry out extracellular recordings, a borosilicate pipette filled with *Xenopus* Ringer was targeted to the olfactory glomerulus showing GFP fluorescence. Signals were acquired using a Geneclamp 500A amplifier (Molecular Devices, San Jose, CA, USA) and digitized at 10 kHz using a National Instruments NI-USB-6341 DAC board (National Instruments, Austin, TX, USA) controlled by mafPC software (courtesy of MA Xu-Friedman, University at Buffalo, NY, USA). The recorded changes of the LFP were low-pass filtered below 100 Hz and analyzed with Igor Pro 9.0 (WaveMetrics, OR, USA). *Xenopus* tadpoles were euthanized upon completion of recordings by placing them in 0.6% MS-222.

Methionine was chosen as odor stimulus based on its broad capacity to stimulate OSNs in *X. laevis* tadpoles (*Manzini and Schild, 2004*). A puff of 200 µM methionine solution, obtained by diluting a 10 mM stock solution prepared in *Xenopus* Ringer (pH = 7.8), was applied on the ipsilateral olfactory epithelium to the recorded glomerulus. The amino acid was delivered through a 0.25 mm diameter fused silica capillary (World Precision Instruments, Hertfordshire, UK) positioned on the top of a nasal cavity. The timing of application was controlled via a TTL pulse. The characteristic odor-evoked glomerular response was obtained by averaging LFP changes triggered by sequential stimulations applied at 2 min intervals. The local application of antagonists was carried out by transiently applying 20 psi pressure to the pipette holder inlet. Approximately 1 µL of the pipette solution was delivered to the glomerulus.

## Imaging

Quantification of OSNs present in the olfactory epithelium of *Dre.mxn1:GFP* tadpoles was performed in an inverted LSM880 confocal microscope (Zeiss, RRID:SCR_020925) using animals anesthetized in 0.02% MS-222. The number of neurons was estimated from maximal intensity projections in confocal Z-stacks. To selectively eliminate GFP-positive OSNs, the 2Phatal method (*Hill et al., 2017*) was adapted to confocal microscopy. Tadpoles were immersed for 15 min in *Xenopus* water containing 5 µg/mL Hoechst 33342 and anesthetized in 0.02% MS-222. Animals were placed dorsally on a 25 mm glass #1.5 coverslip acting as the bottom of an imaging chamber and transferred to the microscope stage. The right olfactory epithelium, located contralaterally to the recording site, was imaged with a 10× Plan Apo objective, NA = 0.45 (Zeiss). The digital zoom was set to 3.5. Photobleaching was carried out using ZenBlue software (Zeiss) in four to six ROIs containing two or more GFP-positive OSNs. Animals were next returned to water tanks.

The *X. tropicalis* transgenic line *ElasGFP:Tubb2b-GCaMP6s*, where the neuronal β-tubulin promoter drives the expression of the calcium sensor GCaMP6s, was used to visualize the activation of olfactory glomeruli by odorants. As for in vivo electrophysiology, tadpoles were anesthetized in 0.02% MS-222, placed in a recording dish, and transferred to the stage of an upright microscope (Zeiss, Axioexaminer A1). Tadpoles were continuously perfused with *Xenopus* Ringer containing 1 µM d-tubocurarine. Imaging was carried out at 76 Hz using a Maico MEMS confocal unit (Hamamatsu Photonics, Hamamatsu City, Japan). A 40×, 0.75 NA water immersion W-N-Achroplan objective (Zeiss) was used. Odorant stimulation was performed as described for electrophysiology experiments. The timing of methionine application was synchronized with image acquisition (HCImage software, Hamamatsu Photonics; RRID:SCR_015041) using a Master 8 stimulator (AMPI, Jerusalem, Israel; RRID:SCR_018889). In experiments where imaging was coupled to recordings of the LFP, the electrode was targeted to the lateral glomerular cluster. All signals were synchronized using a Master 8 stimulator.

Raw fluorescence image sequences were used to construct $\Delta F/F$ movies according to the relationship $(F-F_0)/F_0$, where $F_0$ corresponded to the basal fluorescence levels obtained during 1 s before stimulation. $\Delta F/F$ sequences were next subsampled by averaging groups of 10 frames to identify putative glomeruli, which were defined as round structures 10–20 µm in diameter that consistently responded to sequential stimulations applied at 2 min intervals. The selected ROIs were transferred to the original raw fluorescence movie to quantify increases in basal intracellular calcium levels. An ROI showing $\Delta F/F$ increases evoked by methionine application to the ipsilateral olfactory epithelium that were ≥3 SDs above baseline levels was considered as a single glomerulus. Calcium imaging of the dorsolateral pallium was carried out as in the olfactory bulb, but using ROIs measuring 30 µm diameter. The temporal response was quantified using Igor Pro 9 software (WaveMetrics, OR, USA).

The production of ROS in the *X. laevis* HyPer-YFP was evaluated after transection of a single olfactory nerve. Tadpoles were imaged in an inverted confocal LSM 900 microscope (Zeiss, RRID:SCR_022263)

using a Plan-Apochromat 20×, 0.8NA objective. Hyper-YFP was excited at 405 nm and 488 nm. ROS levels were estimated from the relationship obtained between HyPer-YFP excited at 488 nm and 405 nm (*Love et al., 2013*).

## Uncaging of Rubi-glutamate

Rubi-glutamate was added to the pipette solution at a final concentration of 300 µM. Procedures were carried out considering the high sensitivity of the caged compound to light (*Fino et al., 2009*). To prevent spontaneous activation of Rubi-glutamate, glass pipettes were coated with beeswax. The recording electrode was positioned above the lateral region of the glomerular layer using dim transmitted light. Next, the GFP-labeled glomerulus was targeted using blue light attenuated with an ND filter (0.5 OD). Approximately 1 µL of the pipette solution was injected in the glomerulus in the absence of light. Upon certifying that the recording of LFP signal was stable for 2 min, a TTL signal opened a shutter (Lambda SC, Sutter Instrument, Novato, CA, USA) for 500 ms to deliver blue light (470±20 nm) through the epifluorescence port. A diaphragm restricted light application to the region targeted by the electrode.

## Histological procedures

*X. tropicalis* tadpoles were anesthetized in 0.02% MS-222, subsequently fixed for immunohistochemistry in 4% PFA during 4–7 days at 4°C and immersed in 30% sucrose. Animals were next embedded in O.C.T. freezing medium (Tissue-Tek, Sakura Finetek, Zoeterwoude, the Netherlands), snap-frozen in isopentane in a Bright Clini-RF rapid freezer, and stored at –80°C until use. Sagittal sections (15 µm thick) were obtained using a cryostat (Leica, Reichert-Jung, Heidelberg, Germany) and mounted on superfrost plus slides (VWR International GmbH, Barcelona). For immunofluorescent staining, the sections were blocked for 2 hr at room temperature with PBS solution containing 0.2% Triton X-100 and 10% NGS. They were next incubated in a moist chamber overnight at 4°C in PBS with 0.2% Triton X-100 and 2% NGS containing anti-tyrosine hydroxylase (mouse monoclonal, Immunostar cat. no. 22941, 1:250; RRID:AB_57226) and anti-GFP (rabbit polyclonal, A6455, Invitrogen, 1:300, RRID:AB_221570). After three washes with PBS, sections were incubated with appropriate secondary antibodies followed by DAPI staining (1:10,000) and mounted with fluoromount (Sigma-Aldrich, Saint Louis, MO, USA). Some sections were obtained from tadpoles whose nasal cavities were injected with CM-DiI (Thermo Fisher Scientific, Walthman, MA, USA, cat. no. C7001) 24 hr before fixation.

## Statistical analysis

For statistical analysis, two-tailed paired and unpaired t-tests were used to evaluate differences between two experimental groups. Comparisons among three or more groups were performed using one-way ANOVA followed by multiple comparison Tukey's HSD test. Average values are expressed as mean ± s.e.m. Statistical analysis was carried out using Igor Pro software (RRID:SCR_000325).

## Acknowledgements

This work was sponsored by the Ministry of Science, Innovation and Universities (MICIU/AEI), grant PID2021-124536NB-I00 (AL), co-funded by the European Regional Development Fund (ERDF), 'a way of making Europe'. The work was also supported by two Whitman Fellowships awarded to AL in 2023 and 2024 (Marine Biological Laboratory, University of Chicago). The authors thank the institutional support from the María de Maeztu Unit of Excellence, Institute of Neurosciences, University of Barcelona, CEX2021-001159-M (Ministry of Science, Innovation and Universities) and the CERCA Program of Generalitat de Catalunya. AL is a Serra Húnter fellow. The authors thank Francisco Ciruela for suggesting the use of raclopride.

# Additional information

## Funding

| Funder | Grant reference number | Author |
|---|---|---|
| Ministerio de Ciencia, Innovación y Universidades | PID2021-124536NB-I00 | Artur Llobet |
| Marine Biological Laboratory | Whitman program | Artur Llobet |
| María de Maeztu Unit of Excellence, Institute of Neurosciences, University of Barcelona | CEX2021-001159-M | |

The funders had no role in study design, data collection and interpretation, or the decision to submit the work for publication.

## Author contributions

Marta Casas, Data curation, Formal analysis, Investigation, Methodology; Beatrice Terni, Investigation; Artur Llobet, Conceptualization, Data curation, Formal analysis, Supervision, Funding acquisition, Investigation, Methodology, Writing – original draft, Writing – review and editing

## Author ORCIDs

Beatrice Terni ⓘ https://orcid.org/0000-0001-9548-1625
Artur Llobet ⓘ https://orcid.org/0000-0001-5797-6782

## Ethics

Ethical procedures were approved by the Institutional Animal Care and Use Committee of the regional government (Generalitat de Catalunya, experimental procedure #10753).

Reviewer #1 (Public review): https://doi.org/10.7554/eLife.107710.3.sa1
Author response https://doi.org/10.7554/eLife.107710.3.sa2

# Additional files

## Supplementary files

MDAR checklist

Source data 1. Datasets for Figures 2C, 3A, D, 4B–D, 5D, F, 7B, E, 8C and 9C.

## Data availability

All data generated or analyzed during this study are included in the manuscript. Source data files have been provided.

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
