## [Editor Report · eLife Assessment]

This manuscript investigates inter-hemispheric interactions in the olfactory system of Xenopus tadpoles. Using a combination of electrophysiology, pharmacology, imaging, and uncaging, the transection of the contralateral nerve is shown to lead to larger odor responses in the un-manipulated hemisphere, and implicates dopamine signaling, likely originating from the lateral pallium, in this process. The study **convincingly** uses a rich and sophisticated array of tools to investigate olfactory coding, and uncovers **valuable** mechanisms of signaling likely to be conserved across vertebrates.

---

## [Referee Report · Reviewer #1 (Public review)]

In this study, the authors investigate responses to methionine in the olfactory system of the Xenopus tadpole. They show that the LFP response is local to the glomerular layer, arises ipsilaterally, and is blocked by pharmacological blockade of AMPA and NMDA receptors, with little modulation during blockade of GABA-A receptors. They then show that this response is translently enlarged following transection of the contralateral olfactory nerve, but not the optic lobe nerve. Measurement of ROS- a marker of inflammation- was not affected by contralateral nerve transection, and LFP expansion was not affected by pharmacological blockade of ROS production. Imaging biased towards presynaptic terminals suggests that the enlargement of the LFP has a presynaptic component. A D2 antagonist increases the LFP size and variability in intact tadpoles, while a GABA-B antagonist does not. Finally, the authors provide anatomical and physiological evidence that the contralateral dopamine signal may arise from the lateral pallium. Overall, I found the array of techniques and approaches applied in this study to be creatively and effectively employed.

---

## [Author Response]

The following is the authors’ response to the original reviews.

**Public Reviews:**

**Reviewer #1 (Public review):**
In this study, the authors investigate LFP responses to methionine in the olfactory system of the Xenopus tadpole. They show that this response is local to the glomerular layer, arises ipsilaterally, and is blocked by pharmacological blockade of AMPA and NMDA receptors, with little modulation during blockade of GABA-A receptors. They then show that this response is translently enlarged following transection of the contralateral olfactory nerve, but not the optic lobe nerve. Measurement of ROS- a marker of inflammation- was not affected by contralateral nerve transection, and LFP expansion was not affected by pharmacological blockade of ROS production. Imaging biased towards presynaptic terminals suggests that the enlargement of the LFP has a presynaptic component. A D2 antagonist increases the LFP size and variability in intact tadpoles, while a GABA-B antagonist does not. On this basis, the authors conclude that the increase driven by contralateral nerve transection is due to DA signaling.Overall, I found the array of techniques and approaches applied in this study to be creatively and effectively employed. However, several of the conclusions made in the Discussion are too strong, given the evidence presented. For example, the authors state that "The observed potentiation was not related to inflammatory mediators associated to inury, because it was caused by a release of the inhibition made by D2 dopamine receptor present in OSN axon terminals." This statement is too strong - the authors have shown that D2 receptors are sufficient to cause an increase in LFP, but not that they are required for the potentiation evoked by nerve transection. The right experiment here would be to get rid of the D2 receptors prior to transection and show that the potentiation is now abolished. In addition, the authors have not shown any data localizing D2 receptors to OSN axon terminals.Similarly, the authors state, "the onset of LFP changes detected in glomeruli is determined by glutamate release from OSNs." Again, the authors have shown that blockade of AMPA/NMDA receptors decreases the LFP, and that uncaging of glutamate can evoke small negative deflections, but not that the intact signal arises from glutamate release from OSNs. The conclusions about the in vivo contribution of this contralateral pathway are also rather speculative. Acute silencing of one hemisphere would likely provide more insight into the moment-to-moment contributions of bilateral signals to those recorded in one hemisphere.

We thank the reviewer for their positive evaluation of our manuscript. We agree with their opinion about the necessity of including new experimental evidence to back up discussion and conclusions

**Recommendations for the authors:**

**Reviewer #1 (Recommendations for the authors):**
This is a creative and careful study, but I felt that the conclusions in the Discussion were too strong. I think these could either be toned down or additional experiments could be done to support the idea that D2 receptors are required for the nerve transection-evoked potentiation, that the source of glutamatergic input is OSNs, and that contralateral interactions are mediated by DA. In particular, I think anatomical stains showing which neurons are carrying the DA signal and whether there is any potentiation of DA release after nerve transection would greatly strengthen the conclusions.

This new version of the manuscript contains two new figures: 6 and 9.

New figure 6 addresses the suggestion of this reviewer and provides anatomical evidence for the distribution of dopaminergic neurons in the olfactory bulb of *X. tropicalis* tadpoles using a tyrosine hydroxylase antibody (mouse monoclonal, Immunostar cat. no. 22941, 1:250; RRID:AB_57226). We identified a discrete neuronal population present in the border between the mitral cell layer and the glomerular layer that resembles the type1 TH+ population described in adult frogs (Boyd and Delaney 2002). TH+ neurons send their processes to innervate olfactory glomeruli and we provide evidence that they contact the GFP lateral glomerulus labelled in Dre.mxn1:GFP *X. tropicalis* tadpoles (Fig. 6C). These results reinforce a modulatory role for dopamine on glomerular neurotransmission. Materials & methods (lines 152-167), results (lines 393-399) and discussion (lines 550-563) have been modified accordingly.

Figure 9 provides new evidence on the interhemispheric connections involved in the potentiation of glomerular responses. We first demonstrate that dorsolateral pallial neurons participate in the processing of olfactory information based on the general consideration that the lateral pallium is an olfactory cortex. We confirmed this possibility by stimulating the olfactory epithelium and recording ipsilateral calcium transients in pallial neurons of *tubb2b:GCaMP6s* tadpoles. We next injured the dorsolateral pallium and 24-48h afterwards we recorded odor-evoked responses in the GFP labelled glomerulus located contralaterally. We observed a ~70% potentiation of responses, which was comparable to the ~75% potentiation obtained by olfactory nerve transection. These results illustrated the involvement of pallial neurons in the control of glomerular output by likely modifying the activity of TH+ neurons. The results (473-506) and discussion (569-576) now include these new results.

Does the contribution of DA signalling change across development? I think this would be helpful to interpret the results and relatively straightforward to do: apply raclopride at different developmental stages and measure how much potentiation occurs at each stage.

This is indeed an interesting point, but conducting a comprehensive study of dopamine release throughout development would require a substantial amount of work and delay the publication of this paper. To perform these experiments, we should first implement new technical approaches, such as successfully injuring young tadpoles or recording from late premetamorphic stages. We believe that the proposed experiments could define a new line of arguments rather than complement the present work. Nonetheless, we acknowledge the suggestion of this reviewer.

In this new version, we provide strong evidence for dopamine release in the glomerular layer, and a key question that arises is the nature of TH+ positive neurons. Recent findings obtained in mice show that there are five different types of dopaminergic interneurons present in the olfactory bulb (Kosaka, Pignatelli, and Kosaka 2020), and important functional differences exist between axon-bearing and anaxonic neurons (Dorrego-Rivas et al. 2025). This evidence suggests a key role for development. A completely new study based on transgenic *X. tropicalis* displaying labeled TH+ neurons could bring together development, anatomy, and physiology to gain an understanding of how dopaminergic signaling shapes glomerular function.

In addition, there are several places where showing additional raw data in the figures and carefully quantifying variability would be helpful. For example, in Figure 3B, the authors should show equivalent raw traces from intact and transected tadpoles. In Figure 5D, it would be helpful to show raw traces for LFP equivalent to what is shown for presynaptic imaging in Figure 5E. In Figures 6E-F, it would be helpful to show raw traces.

Thank you for this suggestion. The examples have been added to the figure panels.

I found the last experiment with photobleaching somewhat inconclusive, and I am not sure what it adds to the study as presently written. Line 418: Please quantify how many OSNs remained. Line 423: What is the hypothesis for the source of variability?

The goal of this experiment is to investigate the participation of chemotopy in the potentiation induced by contralateral injury. The elimination of 30-50% of topographically related OSNs did not alter contralateral glomerular responses. This evidence suggests that chemotopy was not relevant to the gain of function observed ; however, we cannot completely rule out a certain topographical contribution, as it was not possible to completely silence all inputs of the studied glomerulus. We now link these findings to the likely innervation of several glomeruli by TH+ neurons, which suggests the absence of a one-to-one glomerulus relationship. LFP amplitudes and their variance are now illustrated in box plots to highlight the absence of significant differences. Lines (457-471).

An increase in the variance among the recordings obtained is a consistent empirical observation. Although it is a hallmark of the potentiation recorded, we cannot provide a mechanistic explanation. Considering that neurotransmitter release from OSN axon terminals is normally inhibited by dopamine, we hypothesize that disinhibition drives an increase in release probability , leading to larger variations in glutamate release. Such variations could be reflected in the amplitude of LFP negativities.

It would be helpful to include a measurement of LFP over time so we have some idea of how stable the odor delivery is.

The amplitude of LFP responses was stable for >30 min. Figure 3B shows recordings obtained during 30 min and new Figure 7F over 42 min. We believe that these examples illustrate that the amplitude, as well as kinetics of the responses obtained were consistent over the period studied.

Line 227: Small upward deflection - could this be an electrical artifact? Can you run the stimulus delivery with no odor (say, with water) to see if you get the same signal?

We do not know the precise source of this upward deflection. It is not an electrical artifact related to stimulation, which is sometimes evident (Fig 7A, methionine application). When present, it occurs after the activation of OSNs. One possibility is that the deflection originates in the layer of nerve fibers reflecting some aspect related to the conduction of APs and the relative position of the electrode. Interestingly, some recordings of LFP responses at the level of glomeruli carried out in rats also show a positive deflection (see Figs. 1B, 2A, 3B in Lecoq, Tiret, and Charpak 2009), thus suggesting it is an intrinsic characteristic of this type of recordings.

Line 237-239: I wasn't clear from the text whether this was a variation due to development, to transection, or natural variability.

We now indicate that the relationship reflects normal development (lines 261-264).

Line 521: N-type VGCCs: can these be targeted with pharmacology to strengthen the argument?

We acknowledge this suggestion but we have not carried out these experiments as we believe that the interpretation could be complex due to the high density of synapses present in glomeruli and the likely involvement of other types of VGCCs in neurotransmitter release.

Small issues:(1) Line 190-196: Some of this could potentially be moved to the Discussion section.

These are some arguments to defend the validity of our experimental approach to record the response of the lateral glomerulus labeled by GFP. If we move them to the discussion, the information related to the spatial extent of our recordings would be split between results and discussion. We believe that the current format of the paper allows to focus the discussion on the interpretation of the results obtained.

(2) Line 268: exponential recover phase.

Thanks. Corrected.

(3) Line 278: affected to -> arises from

Thanks. Corrected.

(4) Line 282: affect to -> can affect.

Thanks. Corrected.

(5) Line 403: 2Phatal technique: Please state briefly what this is

It is now indicated: two-photon chemical apoptotic targeted ablation (2Phatal).

NOTE:

During the revision of this manuscript we realized that Figures 3C and 4B indicated mean ± SD. The panels have been amended to show mean ± s.e.m.

References

Boyd, J. D., and K. R. Delaney. 2002. "Tyrosine hydroxylase-immunoreactive interneurons in the olfactory bulb of the frogs *Rana pipiens* and *Xenopus laevis*." J Comp Neurol 454 (1):42-57. doi: 10.1002/cne.10428.

Dorrego-Rivas, A., D. J. Byrne, Y. Liu, M. Cheah, C. Arslan, M. Lipovsek, M. C. Ford, and M. S. Grubb. 2025. "Strikingly different neurotransmitter release strategies in dopaminergic subclasses." Elife 14. doi: 10.7554/eLife.105271.

Kosaka, T., A. Pignatelli, and K. Kosaka. 2020. "Heterogeneity of tyrosine hydroxylase expressing neurons in the main olfactory bulb of the mouse." Neurosci Res 157:15-33. doi: 10.1016/j.neures.2019.10.004.

Lecoq, J., P. Tiret, and S. Charpak. 2009. "Peripheral adaptation codes for high odor concentration in glomeruli." J Neurosci 29 (10):3067-72. doi: 10.1523/JNEUROSCI.6187-08.2009.